# Long-Term Hepatitis B Virus Infection Induces Cytopathic Effects in Primary Human Hepatocytes, and Can Be Partially Reversed by Antiviral Therapy

Wenjing Zai,[a] Kongying Hu,[a] Jianyu Ye,[a] Jiahui Ding,[a] Chao Huang,[a] Yaming Li,[a] Zhong Fang,[c] Min Wu,[c] Cong Wang,[c] Jieliang Chen,[a,b] Zhenghong Yuan[a,b]

[a]Key Laboratory of Medical Molecular Virology (MOE/NHC/CAMS), School of Basic Medical Sciences, Shanghai Medical College, Fudan University, Shanghai, China
[b]Research Unit of Cure of Chronic Hepatitis B Virus Infection, Chinese Academy of Medical Sciences, Shanghai, China
[c]Shanghai Public Health Clinical Center, Fudan University, Shanghai, China

**ABSTRACT** Chronic infection of hepatitis B virus (HBV) remains a major health burden worldwide. While the immune response has been recognized to play crucial roles in HBV pathogenesis, the direct cytopathic effects of HBV infection and replication on host hepatocytes and the HBV-host interactions are only partially defined due to limited culture systems. Here, based on our recently developed 5 chemical-cultured primary human hepatocytes (5C-PHHs) model that supports long-term HBV infection, we performed multiplexed quantitative analysis of temporal changes of host proteome and transcriptome on PHHs infected by HBV for up to 4 weeks. We showed that metabolic-, complement-, cytoskeleton-, mitochondrial-, and oxidation-related pathways were modulated at transcriptional or posttranscriptional levels during long-term HBV infection, which led to cytopathic effects and could be partially rescued by early, rather than late, nucleot(s)ide analog (NA) administration and could be significantly relieved by blocking viral antigens with RNA interference (RNAi). Overexpression screening of the dysregulated proteins identified a series of host factors that may contribute to pro- or anti-HBV responses of the infected hepatocytes. In conclusion, our results suggest that long-term HBV infection in primary human hepatocytes leads to cytopathic effects through remodeling the proteome and transcriptome and early antiviral treatment may reduce the extent of such effects, indicating a role of virological factors in HBV pathogenesis and a potential benefit of early administration of antiviral treatment.

**IMPORTANCE** Global temporal quantitative proteomic and transcriptomic analysis using long-term hepatitis B virus (HBV)-infected primary human hepatocytes uncovered extensive remodeling of the host proteome and transcriptome and revealed cytopathic effects of long-term viral replication. Metabolic-, complement-, cytoskeleton-, mitochondrial-, and oxidation-related pathways were modulated at transcriptional or posttranscriptional levels, which could be partially rescued by early, rather than late, NA therapy and could be relieved by blocking viral antigens with RNAi. Overexpression screening identified a series of pro- or anti-HBV host factors. These data have deepened the understanding of the mechanisms of viral pathogenesis and HBV-host interactions in hepatocytes, with implications for therapeutic intervention.

**KEYWORDS** long-term HBV infection, quantitative temporal proteomics, cytopathic effects, nucleot(s)ide analogues, viral-host interaction

Address correspondence to Jieliang Chen, jieliangchen@fudan.edu.cn, or Zhenghong Yuan, zhyuan@shmu.edu.cn.

The authors declare no conflict of interest.

Hepatitis B virus (HBV) infection remains a worldwide public health threat, with approximately 250 million individuals chronically infected and at risk of developing a wide spectrum of liver diseases, including hepatitis, fibrosis, cirrhosis, and hepatocellular carcinoma

(HCC) (1). Current clinical therapeutics, including interferon alpha (IFN-$\alpha$) and nucleot(s)ide analog (NA) treatment can control, but not cure, the disease (2). HBV is a hepatotropic virus with strict specificity for human hepatocytes. The development of liver diseases during HBV infection is thought to be a result of complicated viral-host interactions. The immune response to HBV infection has been recognized to be critical for viral pathogenesis (3–5). However, it remains uncertain how HBV infection and replication in host hepatocytes contribute to viral pathogenesis, and the mechanisms of HBV-host interactions are only partially defined (6–9). A deeper understanding of the mechanisms of viral persistence and pathogenesis in hepatocytes, and identification of related host factors, may provide new insights into potential antiviral targets.

The uncertain physiological relevance of the traditional hepatoma cell line-based experimental models could be a reason for the knowledge gaps mentioned above (6, 10, 11). In recent years, innovative model systems and new systems biology methods have opened the possibility to investigate the complex host-viral interplay in HBV-infected hepatocytes (12–14). Primary human hepatocytes (PHHs) are considered the most physiologically relevant culture system for HBV infection *in vitro*, as they most closely represent the human hepatocytes in the liver (15). However, due to the limited viability and unstable functional maintenance of PHHs *in vitro*, previous studies of the proteome and transcriptome of HBV-infected cells have been restricted to relatively early time points of infection, which hampered comprehensive analysis of the complex network of host-viral interaction (16, 17). The newly established 5 chemical-cultured primary human hepatocytes (5C-PHHs), which could support long-term culture of PHHs *in vitro* and maintain the differentiation status and transcriptional characteristics of hepatocytes, enable highly efficient HBV infection and replication, thus providing a platform to investigate dynamic network relationships between HBV and hepatocytes from mutual antagonism to stable infection (18). Quantitative temporal viromics (QTV) is a method to enable systematic analysis of temporal changes in host and viral proteins throughout the course of viral infection (19–21). By this means, global quantitative profiling of long-term HBV-infected PHHs could provide a comprehensive understanding of HBV-driven cellular reprogramming, with or without antiviral therapeutics interruption, and give novel insights into the pathogenesis of HBV persistent infection.

Here, multiplexed labeling-based system-level quantitative temporal proteomics and parallel transcriptomics were performed in HBV-infected, long-term cultured 5C-PHHs to provide dynamic insights into how HBV orchestrates global expression of host proteins and transcripts during establishment of persistent HBV infection, with or without NA/RNA interference (RNAi) intervention. Functional pathway enrichment analysis of the dysregulated proteins enabled an in-depth understanding of key aspects of HBV persistence and pathogenesis, mainly involved in metabolism-, complement-, cytoskeleton-, and oxidation- related pathways, which could be partially rescued by early NA therapy and relieved by RNAi. Furthermore, overexpression screening identified a series of host factors required for HBV transcription or antagonism among the dysregulated proteins.

## RESULTS

**Temporal profiling of long-term HBV-infected hepatocytes.** The 5C-PHHs (18) were infected with HBV particles collected from HepAD38 supernatants, and long-term HBV infection was successfully achieved over at least 4 weeks, as evidenced by the detection of secreted HBV antigens (HBsAg and HBeAg) and intracellular HBV RNAs, HBc, and covalently closed circular DNA (cccDNA) (Fig. 1A to D; Fig. S1A to C in the supplemental material). Greater than 80% infection efficiency was achieved in this culture system, supported by the result of immunofluorescence staining with anti-HBc antibody (Fig. S1B), as we previously observed (18, 22–24). Notably, the expression levels of HBV antigens and HBV RNAs were relatively low within 2 days postinfection (dpi), which we defined as "acute phase," accumulated up to 10 dpi (defined as "adaptive phase"), and then entered the platform period (10 to 28 dpi, defined as "stable phase"), the kinetics of which could be divided into three different stages. Similarly, the intracellular cccDNA and

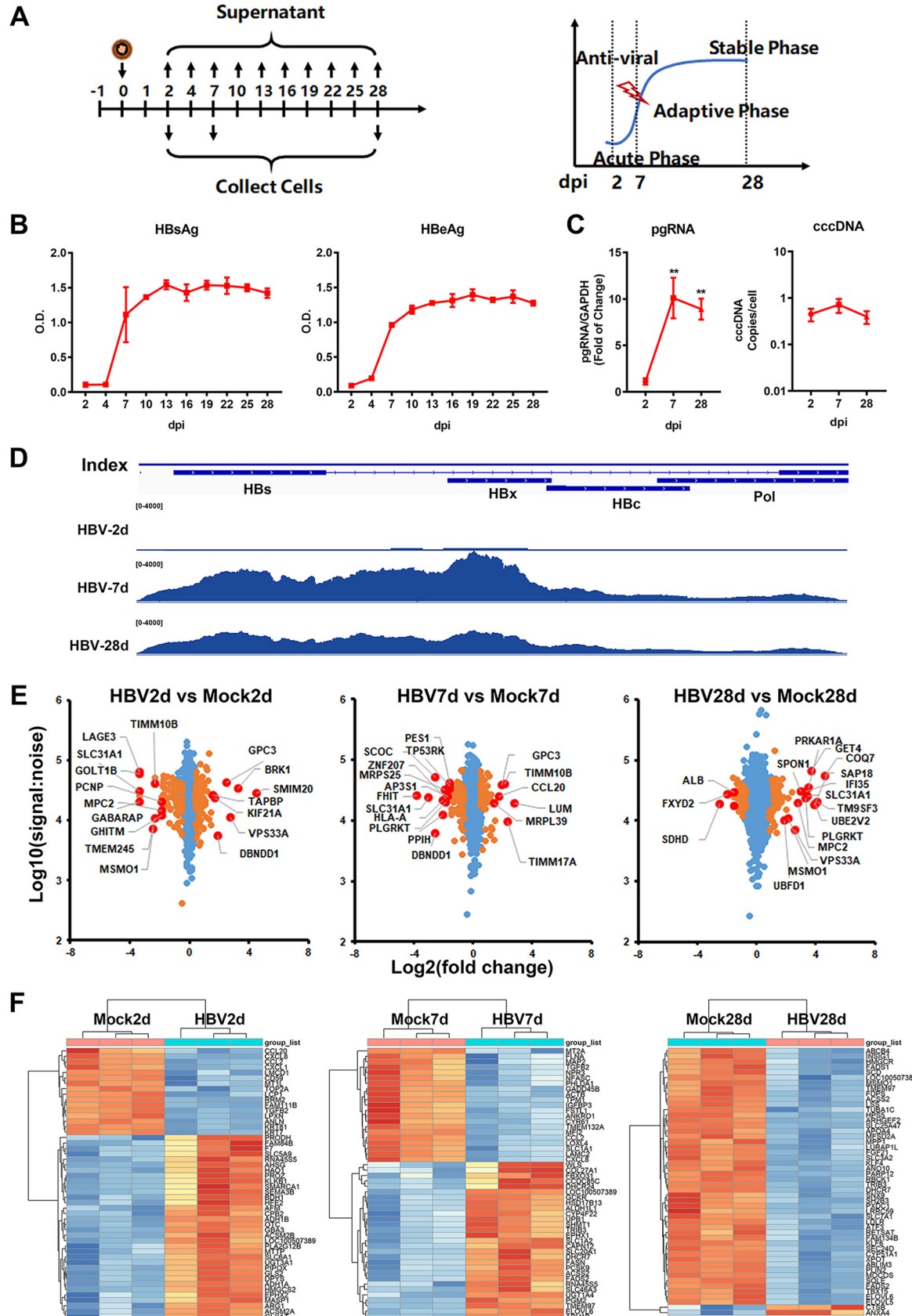

**FIG 1** Temporal profiling of long-term HBV-infected hepatocytes. (A) 5C-PHHs were infected with HBV at an MOI of 200. (B) Extracellular HBsAg and HBeAg were evaluated every 3 days and analyzed by qPCR with specific primers. (C) Intracellular HBV pgRNA

HBc levels (Fig. 1C) were initially accumulated and maintained at high levels for long periods. The gene expression profiles of liver-specific drug-metabolizing enzymes were analyzed, and the expression levels of hepatic surrogate functional markers and hepatic transcriptional factors in 5C-PHHs were determined. The results together showed that the 5C condition effectively supported the expression of hepatic functional genes and maintained the characteristics of human hepatocytes over the long-term culture durations (Fig. S1D and E).

We harvested mock- and HBV-infected 5C-PHHs to quantify kinetics of changes in protein expression by multiplexed labeling and tandem mass spectrometry at 2, 7, and 28 dpi (Fig. S2A to D). A statistical analysis method based on the mass spectrum peptide peak intensities and fold changes between cell populations revealed the top and bottom dysregulated proteins (Fig. 1E). In parallel, the corresponding cellular RNA samples in triplicate were analyzed by RNA sequencing (RNA-seq), and the top dysregulated genes were shown in heatmaps (Fig. 1F; Fig. S2E). Thus, a comprehensive proteomic and transcriptomic data set describing the global changes in protein and RNA contents of cells with HBV long-term infection were generated, with nearly 5,000 human and 1 viral protein (HBc) and >20,000 host and viral transcripts quantified. Particularly, the changes of protein profiles in response to long-term HBV infection (28 dpi) were analyzed in triplicate for more details. The complete interactive data set is shown in Table S1, enabling generation of temporal profiles of any quantified proteins of interest.

**Cytopathic effects caused by long-term HBV infection.** Wide-scale proteome and transcriptome remodeling by HBV infection suggest a model in which HBV manipulates diverse cellular proteins and pathways for viral replication and simultaneously results in cumulative effects on cellular function and phenotypes. Multiple host pathways related to cellular metabolic remodeling, cell cycle arrest, and transcriptional misregulation in cancer were observed by transcriptomics to be dysregulated during HBV infection (Fig. S3A to C). Functional enrichment analysis by the Database of Annotation, Visualization, and Integrated Discovery (DAVID) software package highlighted that interaction between viral and host proteins were involved in several complexes and biological pathways, including metabolic pathways, complement and coagulation cascade, actin cytoskeleton organization, antigen processing and presentation, and oxidative reduction processes (Fig. 2A; Fig. S3D). Components of each enriched cluster are shown in Table S2.

A further ingenuity pathway analysis (IPA) was applied, and the results showed that the major toxicity and pathogenesis of HBV long-term infection were the inducing of mitochondrial dysfunction (Fig. 2B). Ultrastructure analysis by transmission electron microscopy (TEM) displayed that long-term HBV infection led to disrupted cytoskeleton, cytoplasm vacuolation, and abnormal mitochondrial morphologies, and it lost its regular tubular or round shape and organized mitochondrial cristae and exhibited a signature of abnormal membrane integrity and swollen (Fig. 2C). Immunofluorescent analysis confirmed the abnormal mitochondrial morphology and reactive oxygen species (ROS) accumulation in infected 5C-PHHs (Fig. 2D). The fluorescent intensities of retained JC-1 aggregates (red) in mock-infected cells indicated normal mitochondrial membrane potential (MMP), while long-term HBV infection led to a significant decrease of JC-1 aggregates, suggesting loss of MMP (Fig. 2E). Notably, we did not observe an enrichment of innate immunity or interferon-related signaling pathways (Fig. 2; Fig. S3), supporting the notion that HBV triggers little innate immune response in hepatocytes during its infection and replication (6).

**FIG 1** Legend (Continued)
and cccDNA were extracted at indicated time points and analyzed by qPCR with specific primers. (D) Distribution of HBV transcripts along the HBV genome were quantified by nucleotide mapping. Read density for each track was represented by height on the y axis. (E) Mock- and HBV-infected cells were collected at 2, 7, and 28 dpi, and then proteomic analysis was applied. Scatterplots displayed pairwise comparisons between HBV- and mock-infected cells; each point represents a single protein. Benjamini-Hochberg corrected significance was utilized to estimate $P$ values. Orange dots, $P < 0.01$; red dots, top and bottom deregulated proteins. (F) Parallel samples were applied to RNA-seq analysis. Heatmaps displaying top 50 dysregulated genes (left panel, 2 dpi; middle panel, 7 dpi; right panel, 28 dpi).

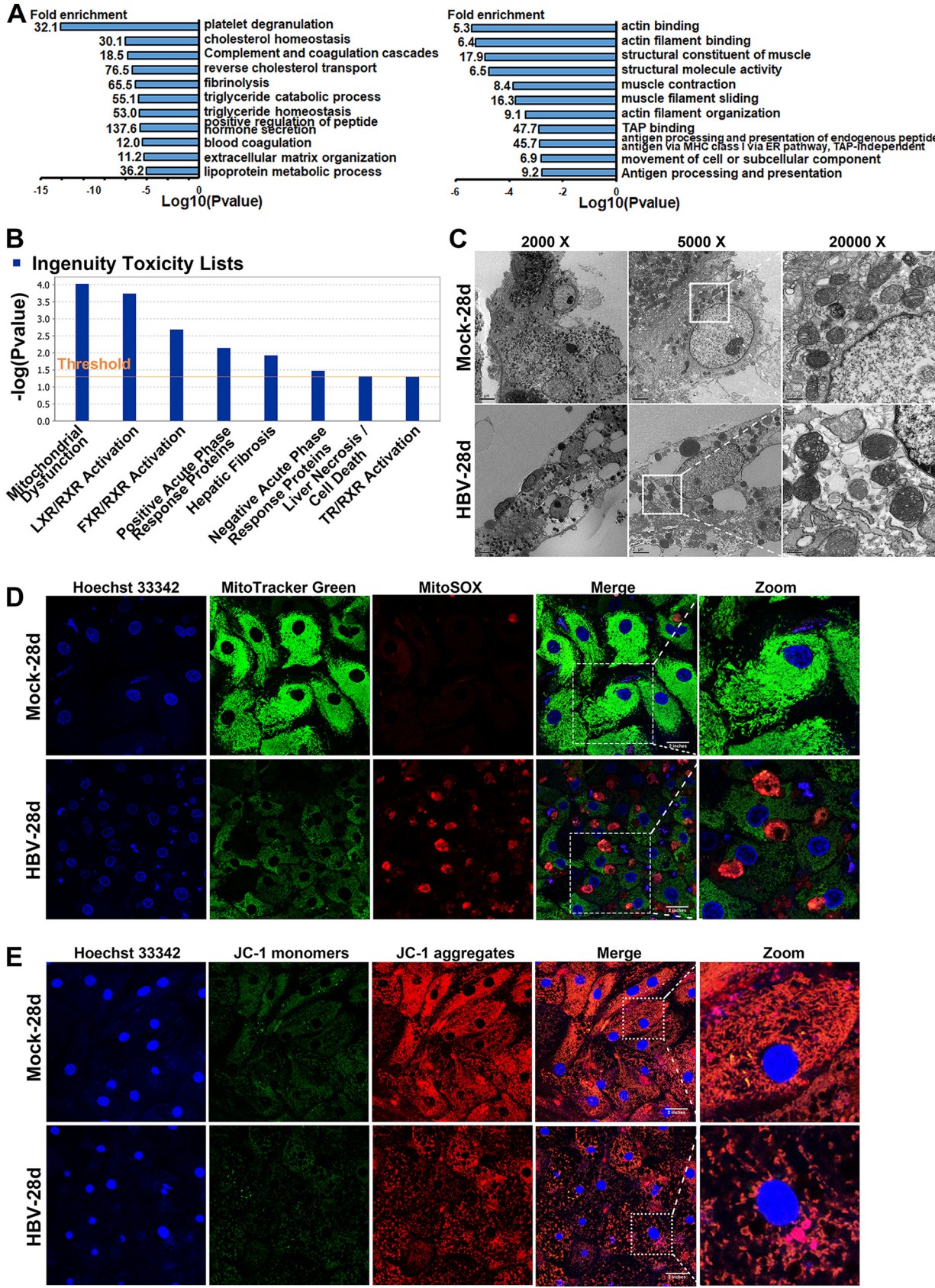

**FIG 2** Cytopathic effects caused by long-term HBV infection. (A) 5C-PHHs were infected with HBV (MOI of 200), and samples were collected at 28 dpi and proteomics analysis applied. DAVID analysis of upregulated (left) and downregulated (right) proteins (>1.5-fold change and

**Transcriptional and posttranscriptional regulation of protein expression.** To interrogate whether protein expression was determined by corresponding mRNA levels or regulated at the posttranscriptional level, we integrated protein and RNA data sets for comparing protein and transcript abundance over time. The k-means method was utilized for clustering proteins into specific numbers of clusters based on the similarity of expression kinetics, and the results showed that there were at least eight different patterns of expression modes, which exhibited as early or later transcriptional up- or downregulation, initially degradation or degradation with time (Fig. 3A; Fig. S4A; also see Table S3). Among them, cluster 3 included proteins that were transcriptionally downregulated, such as AGFG1; cluster 4 included transcriptionally upregulated proteins, like HIBADH; cluster 6 and 7 included proteins that were initially downregulated at the transcriptional level and then upregulated; cluster 5 and 8 included proteins that were degraded over time via posttranscriptional ways, which were also enriched in proteins defined as "protein degradation" (for proteins, downregulation, >1.5-fold change, $P$ value < 0.05; for RNA, unchanged or upregulation, >1-fold change) (Fig. 3B and C). Expression levels of proteins defined as "degraded" were also backtracked and displayed in heatmaps (Fig. 3D).

We also analyzed the overlap of proteomic hits with annotated UniProt keywords of "antiviral defense" and "innate immunity" and found a slight enrichment of these proteins in cluster 8 that tend to be degraded by viral infection, including cytosolic sensors like MAVS, DHX36, and DDX3 and antiviral restriction factors like TRIM25, TRIM38, and so on (Fig. S4B). To further identify biological functions of each cluster in an unbiased fashion, we used DAVID software to determine gene ontology (GO) "molecular function" and "biological process" annotations and Kyoto Encyclopedia of Genes and Genomes (KEGG) pathway enrichment (25). Among transcriptionally upregulated proteins and cluster 4, multiple metabolic terms and oxidation-reduction processes were identified, while pathways enriched in transcriptional downregulation and cluster 3 included lysosome and negative regulation of growth. We also identified that pathways related to rRNA process, actin fragment organization, and mRNA splicing via spliceosome were enriched in protein degradation and cluster 8 (Fig. S4C). Examples of related pathway proteins were displayed in bar plots, and high accordance was observed among different omics (Fig. S4D).

**Nucleot(s)ide analogues treatment partially rescued the HBV infection-mediated dysregulation of protein expression.** As NAs are one of the first-line treatments for chronic hepatitis B (CHB), we further analyzed the changes of cell phenotypes and functions in the presence of NAs. We used entecavir (ETV) to treat the cells (at 100 nM) right away following HBV infection, and samples were collected at 7 and 28 dpi and then applied to proteomics and transcriptomics analysis (Fig. 4A). Viral kinetics analysis showed that early ETV treatment significantly inhibited HBV DNA, HBsAg, and HBeAg levels in the supernatant, as well as reduced intracellular HBV RNAs, as indicated by RNA-seq analysis (Fig. 4B and C; Fig. S2C). The proteomic results showed that early administration of ETV partially restored expression of several dysregulated proteins by HBV infection according to the "stringent criteria" (>1.5-fold change, rescue ratio > 1.5, rescue ratio $P$ < 0.05), among which degraded proteins that associated with endomembrane system accounted for the largest proportion (Fig. 4D and E; Fig. S5A and B), while the degraded proteins that could not be rescued by early ETV administration were mainly related to protein biosynthesis and secretion (Fig. 4F). Meanwhile, it was found that early ETV treatment significantly reduced the oxidative stress and mitochondrial dysfunction in HBV-infected cells and improved the pathway of antigen processing and presentation, according to the "sensitive criteria" (>1-fold change, rescue ratio > 1.5, rescue ratio $P$ < 0.05) (Fig. S5C and D).

**FIG 2** Legend (Continued)
$P$ < 0.05) against a background of all proteins quantified in the proteomics. Components of all clusters are shown in Table S2 in the supplemental material (B) Integrity toxic analysis of all dysregulated proteins by IPA analysis. (C) 5C-PHHs were infected with HBV at an MOI of 200 for 28 days. Representative TEM photographs showing abnormal mitochondria ultrastructure after long-term HBV infection in hepatocytes. (D) Immunofluorescent analysis of Hoechst 33342 (blue), MitoTracker Green (green), and MitoSOX (red) stain in mock- and HBV-infected 5C-PHH cells. (E) Confocal photographs of Hoechst 33342 (blue) and JC-1 (monomers, green; aggregates, red) stain in mock- and HBV-infected 5C-PHH cells, as indicated.

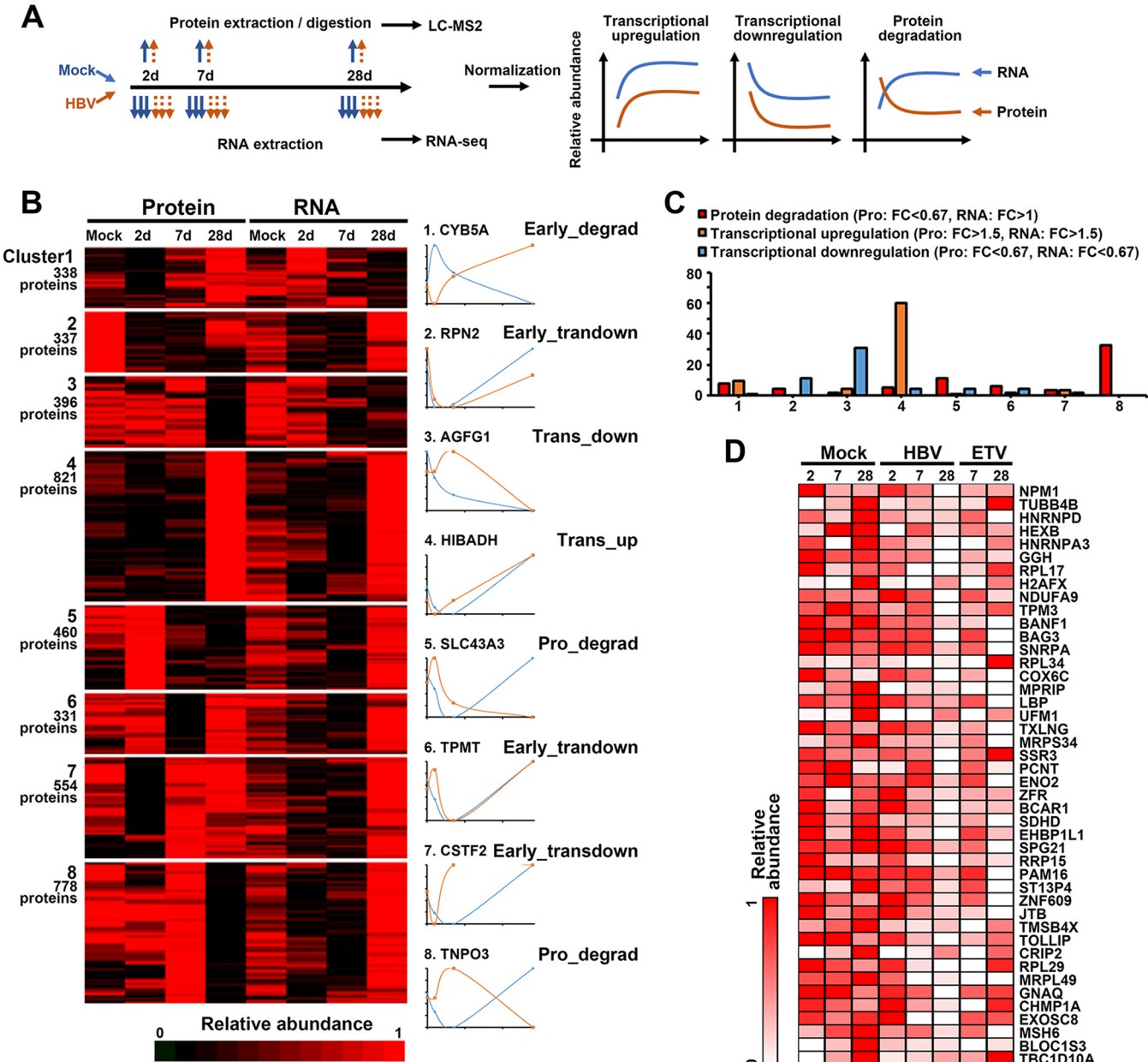

**FIG 3** Transcriptional and posttranscriptional regulation of protein expression. (A) Schematics of experiment workflow. (B) K-means-based hierarchical cluster analysis of proteins and transcripts to identify global mechanisms of protein regulation by HBV infection. Right panels show examples of each cluster. (C) Numbers of proteins from protein degradation, transcriptional upregulation, and transcriptional downregulation shortlists according to relative criteria appearing in each cluster. (D) Proteins from protein degradation appearing in cluster 5 and cluster 8 are displayed.

We additionally quantified proteomic changes in cells treated with NAs only during the stable phase (10 to 28 dpi) (Fig. 5A). By adding ETV (1 $\mu$M) or tenofovir (TDF; 5 $\mu$M), while viral antigens in the supernatant and intracellular cccDNA levels remained unchanged, HBV DNA levels were significantly reduced (Fig. 5B to D). This approach theologically inhibited viral replication and *de novo* infection. In this circumstance, changes induced by blocked HBV replication reproduced the completed HBV life cycle-dependent proteome remodeling, with high degrees of correlation (ETV, $r^2 = 0.7986$; TDF, $r^2 = 0.7863$) (Fig. 5E and F). These data indicated that it was HBV cccDNA transcription and the following antigen expression, but not viral replication, that mainly accounted for the cellular remodeling of infected hepatocytes. RNA-seq analysis further supported the conclusion that early treatment with NAs could partially relieve the abnormal pathways induced by long-term HBV infection and thus reduce the HBV-induced cytopathic effects (Fig. 5G and H).

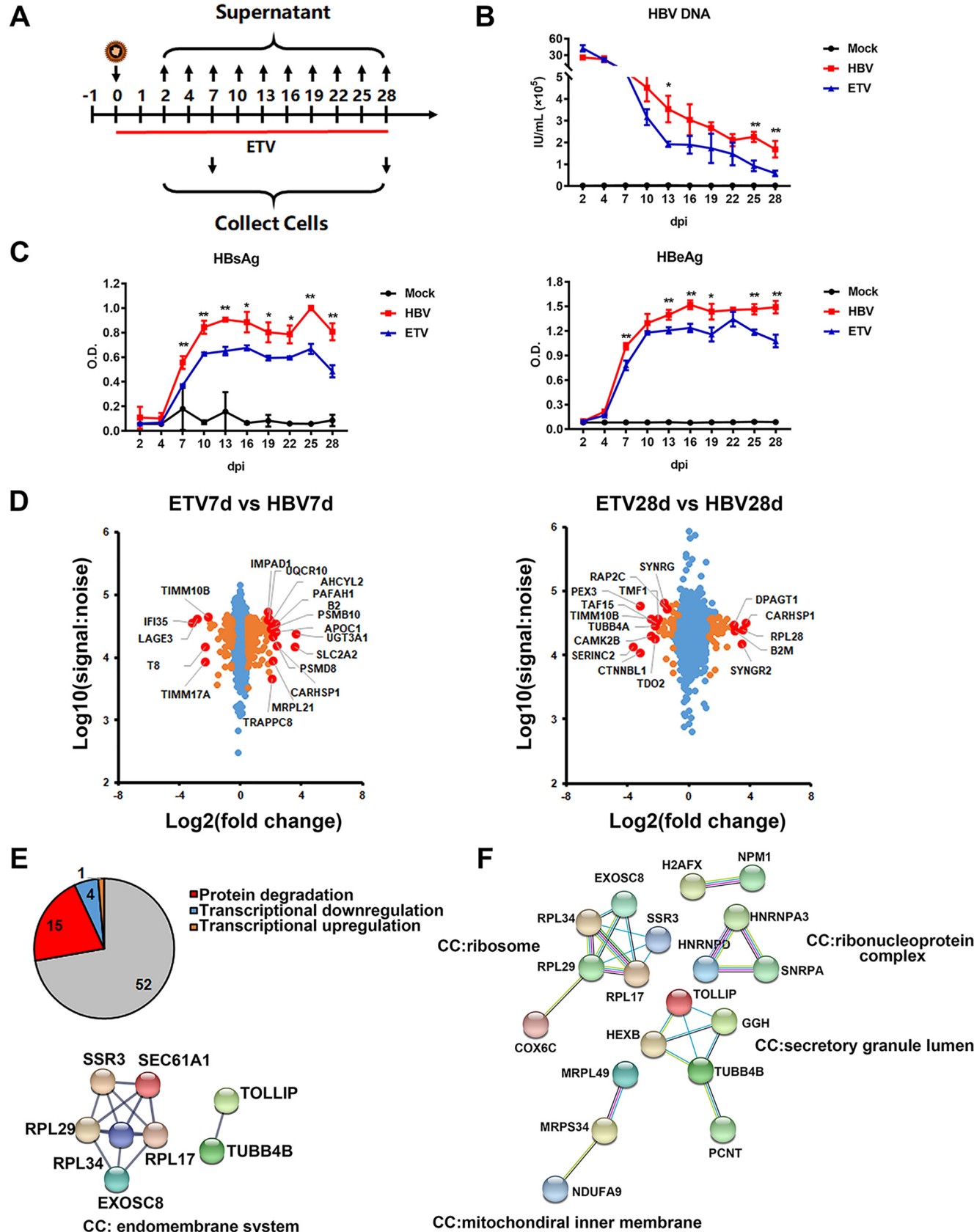

**FIG 4** Identification of proteins dysregulated by HBV while recovered by nucleot(s)ide analogues. (A) 5C-PHH cells infected with HBV (MOI of 200) were treated with or without entecavir (100 nM) at 0 dpi for 28 days. (B and C) Supernatants were collected every 3 days, and HBV DNA (B) and HBsAg and

**Blocking HBV antigen by RNAi treatment significantly relieved HBV-induced host remodeling.** Since viral replication was speculated not to be responsible for HBV-mediated cytopathy, we further inquired whether blocking viral antigen expression by RNAi could reverse long-term HBV infection-induced cellular pathology and host remodeling. We utilized a combination of two small interfering RNA (siRNAs) (siHBV, 100 nM) to degrade all kinds of HBV RNAs (targeting both HBs and HBx sequences). The siHBV was transfected by commercial reagents at 7 days post-HBV infection, the viral antigens were significantly downregulated both 3 and 6 days later, and the intracellular HBV RNA levels were efficiently reduced as evidenced by nucleotide mapping (Fig. 6A to C). Proteomic and transcriptomic analysis was then applied to identify genes or pathways that could be rescued by RNAi (Table S4). According to the stringent criteria in Fig. S5A, we found that genes related to virus infection responses (herpes simplex virus 1 infection), cancer-related signaling (PI3K-Akt/Ras signaling pathway), and cytoskeleton (regulation of actin cytoskeleton), as well as genes associated with cell cycle and cellular adhesion molecules, were reversed by RNAi treatment in the transcriptomic data set (Fig. 6D).

Similarly, the proteomic data set showed that RNAi therapy rescued some proteins that were dysregulated by HBV infection, with a ratio of ~17.2% (21/122) in downregulated and ~5.4% (7/130) in upregulated proteins according to the stringent criteria (Fig. S6A to C). DAVID analysis found that the rescued proteins were enriched in pathways related to complement activation and actin monomer that tend to be dysregulated by long-term HBV infection (Fig. S6D). Further analysis according to the sensitive criteria showed that RNAi therapy also enhanced some immune-related pathways in treated cells, such as innate immune response in mucosa, antibacterial humoral response, and defense response to Gram-positive bacterium (Fig. S6E). In addition, immunofluorescent staining showed that RNAi treatment efficiently relieved HBV infection-induced oxidative stress and mitochondrial dysfunction (Fig. 6E). Together, our results indicated that using antiviral agents like RNAi to block HBV antigen expression could benefit infected cells, not only by rescuing dysregulated proteins but also by enhancing host antiviral responses.

**Screening of dysregulated proteins identified pro- and anti-HBV factors.** As noted above, proteins dysregulated by viral infection are likely to be enriched in factors with pro- or antiviral activity. To further identify host factors related to HBV regulation or antagonism, we integrated the time series screen (640 differently expressed genes [DEGs]), long-term screen (188 DEGs), ETV rescue screen (72 DEGs), and RNA and protein screen (206 DEGs). A total of 117 genes were selected as the union of genes selected using the following keywords: nuclear localization, DNA/RNA binding, transcriptional or epigenetic regulation, signal transduction, viral process, and immune response, for further screening and verification (Fig. 7A; also see Table S5). We then constructed overexpression plasmids for candidate genes and used the recombinant cccDNA system as cccDNA regulatory factor screening model and HBsAg and HBeAg as screening indicators. Two rounds of experiment were taken, and the average values were normalized by Z-score. Most genes had little effects on the propagation of HBV, as the levels of HBsAg and HBeAg in the medium were comparable to that of the control, while there were several genes that could significantly disrupt or promote HBV antigen expression. Genes with the most potent up- or downregulation of HBsAg and HBeAg ($|Z\text{-score}| > 1$) were selected for a second round of screening (Fig. 7B). Five host factors, including UBE2V2, EDF1, WDR3, ZFR, and ESRRB, were identified to significantly promote HBV antigen expression, while eight host factors, including PGAM5,

**FIG 4** Legend (Continued)
HBeAg analyses (C) via commercial kits were applied. (D) Scatterplots of protein changes in comparison between values of ETV-treated and the corresponding HBV-infected cells (left, 7 dpi; right, 28 dpi). *P* values were estimated using Benjamini-Hochberg-corrected significance. Orange dots, *P* < 0.01; red dots, top and bottom deregulated proteins. (E) Overlap of proteins identified in ETV rescue screen (stringent criteria; see Fig. S5A in the supplemental material) and RNA and protein screen. STRING was applied to analyze protein correlation that was degraded by HBV infection but rescued by early ETV treatment. (F) STRING analysis of protein-degraded targets that could not be rescued by ETV treatment.

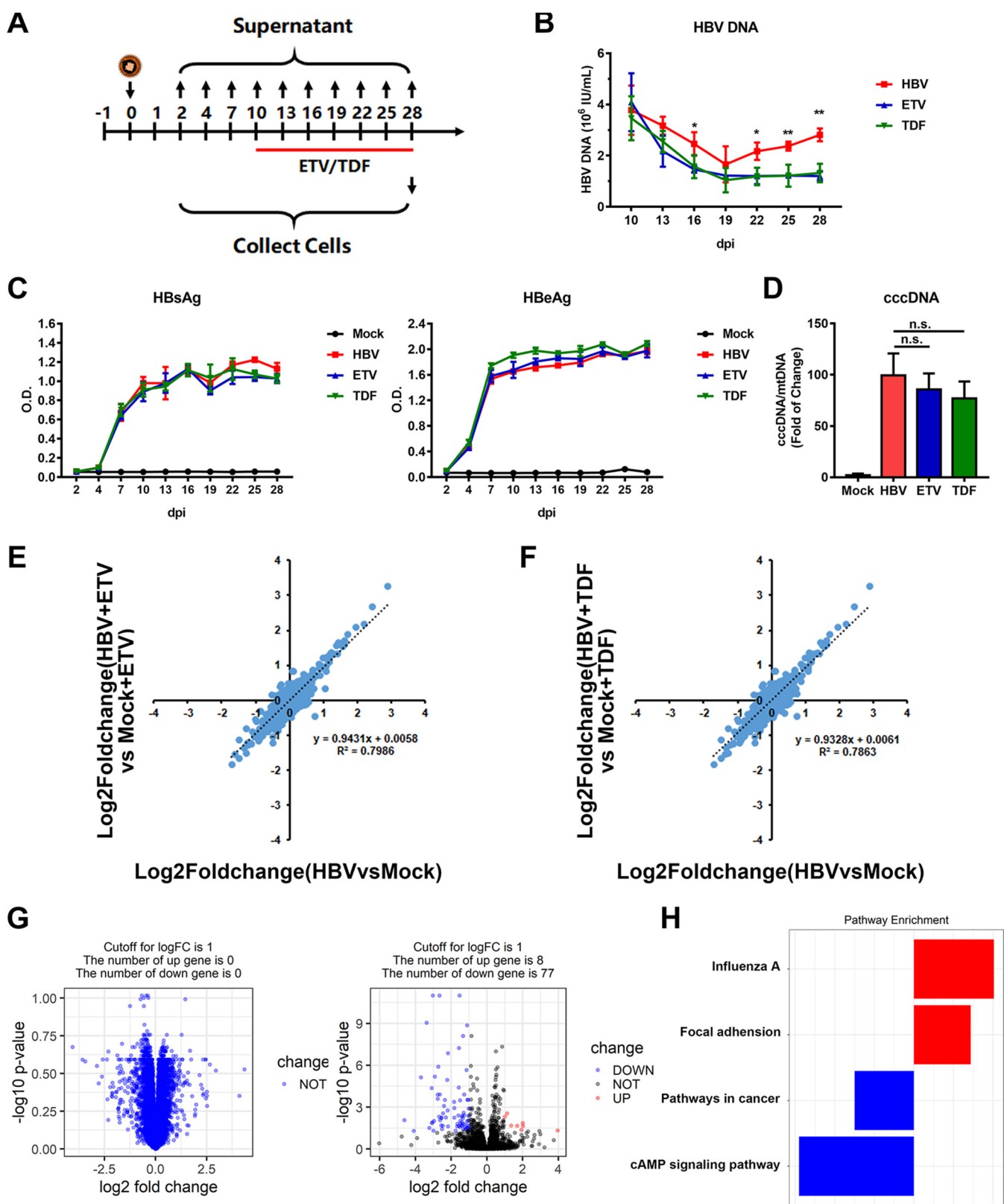

**FIG 5** Late administration of nucleot(s)ide analogues had limited effects on relieving HBV cytotoxicity. (A) 5C-PHH cells were infected with HBV (MOI of 200) and then treated with or without ETV (1 $\mu$M) or TDF (5 $\mu$M) at 10 dpi. (B and C) Supernatants were collected every 3 days and applied to HBV DNA (B) and HBsAg and HBeAg (C) analysis. (D) Intracellular cccDNA was extracted and detected by qPCR with indicated primers. (E) Comparison of ETV-treated or untreated HBV-infected cells with mock-infected cells for 28 days. Values were log$_2$ transformed. (F) Comparison of TDF-treated or untreated HBV-

EXOSC6, CAV1, CAV2, FUBP3, TRIM47, GET4, and HNRNPD, were identified to inhibit HBV antigen secretion (Fig. 7C). The efficacy was also verified in the pHBV1.3-based HBV replication cell model (Fig. S7A). Follow-up studies further confirmed that these genes could inhibit or promote HBV transcription and viral replication in various degrees, as evidenced by decreased or increased intracellular HBV RNAs (including pregenomic RNA [pgRNA] and total RNA) and core particle DNA levels, respectively (Fig. 7D to F). The expression levels of some proteins in the 5C-PHH system were verified to be consistent with the results of proteomics (Fig. S7B). In addition, some of these genes could also regulate cccDNA transcriptional activity, which was calculated as the ratio of pgRNA to cccDNA (Fig. S7C).

## DISCUSSION

The present study provides a comprehensive, unbiased global temporal profiling of HBV-infected human hepatocytes, with implications of mechanisms of viral pathogenesis and viral-host antagonism. The major findings were summarized as follows: (i) long-term HBV infection leads to global proteome and transcriptome remodeling (>5,000 host proteins were quantified, including 1 viral protein [HBc], and >20,000 host and viral transcripts were analyzed), which consists of pathways that include metabolic processes, complement and coagulation cascade, actin cytoskeleton organization, antigen processing and presentation, and oxidative reduction processes; (ii) long-term HBV infection in 5C-PHH leads to cytopathic effects on hepatocytes, as evidenced by dysregulated mitochondrial function and oxidative stress, which could be partially rescued by early, rather than late, NA administration and could be relieved by RNAi; and (iii) overexpression screening of HBV-dysregulated genes identifies a series of host factors that can facilitate or disrupt HBV replication. These data provoke an in-depth understanding of viral-host interplay and reveal related pathways and factors (Fig. 8), thus providing new insights into intervention strategies.

The establishment of HBV stable infection is quite a time-consuming process, which is different from viruses like human cytomegalovirus (HCMV) that could form productive infection within a short time (usually within 24 h) (20, 26). Moreover, unlike viruses such as enterovirus that can cause cytopathic effects (CPE), HBV has long been considered a noncytopathic virus. Ineffective T cell responses, which trigger chronic inflammatory liver injury but not efficient viral eradication, are believed to mainly account for liver fibrosis, cirrhosis, and, finally, hepatocellular carcinoma (27–30). Recent studies indicate that HBV itself might have oncogenic potential that contributes to HCC development via deregulating the cell cycle to render a cellular environment favorable for productive infection and triggering a premalignant phenotype, but focusing on relative early phases (17, 31). The present study is based on the 5C-PHH model, which can support long-term HBV infection for at least 4 weeks. Using this system, we reveal extensive cellular remodeling effects by long-term HBV infection. Specifically, metabolic reprogramming- and oxidation-related changes accumulated with time via transcriptional upregulation mode, while cytoskeleton reorganization-, rRNA/mRNA processing-, and intracellular protein transport-related proteins were posttranscriptionally degraded. We also suggested potential mechanisms of host pathogenesis by long-term viral component load, including mitochondrial dysfunction, cytoskeleton reorganization, metabolic reprogramming, and oxidative stress. The expression levels of genes associated with host remodeling and cytopathology induced by long-term HBV infection were verified by quantitative PCR (qPCR) analysis, and the results showed high consistency with the proteomic and transcriptomic data sets (Fig. S8A in the supplemental material), which further validated the reliability of our conclusions. Further, the possibility of HBV integration was investigated by analyzing the human -HBV chimeric reads within our RNA-seq data

**FIG 5 Legend (Continued)**
infected cells with relative mock-infected cells. (G) Volcano plots displaying pairwise comparisons between early ETV-treated and untreated HBV-infected cells (left, 7 dpi; right, 28 dpi). (H) Gene set enrichment analysis (GSEA) pathway terms enriched among genes rescued by ETV therapy at 28 dpi are displayed.

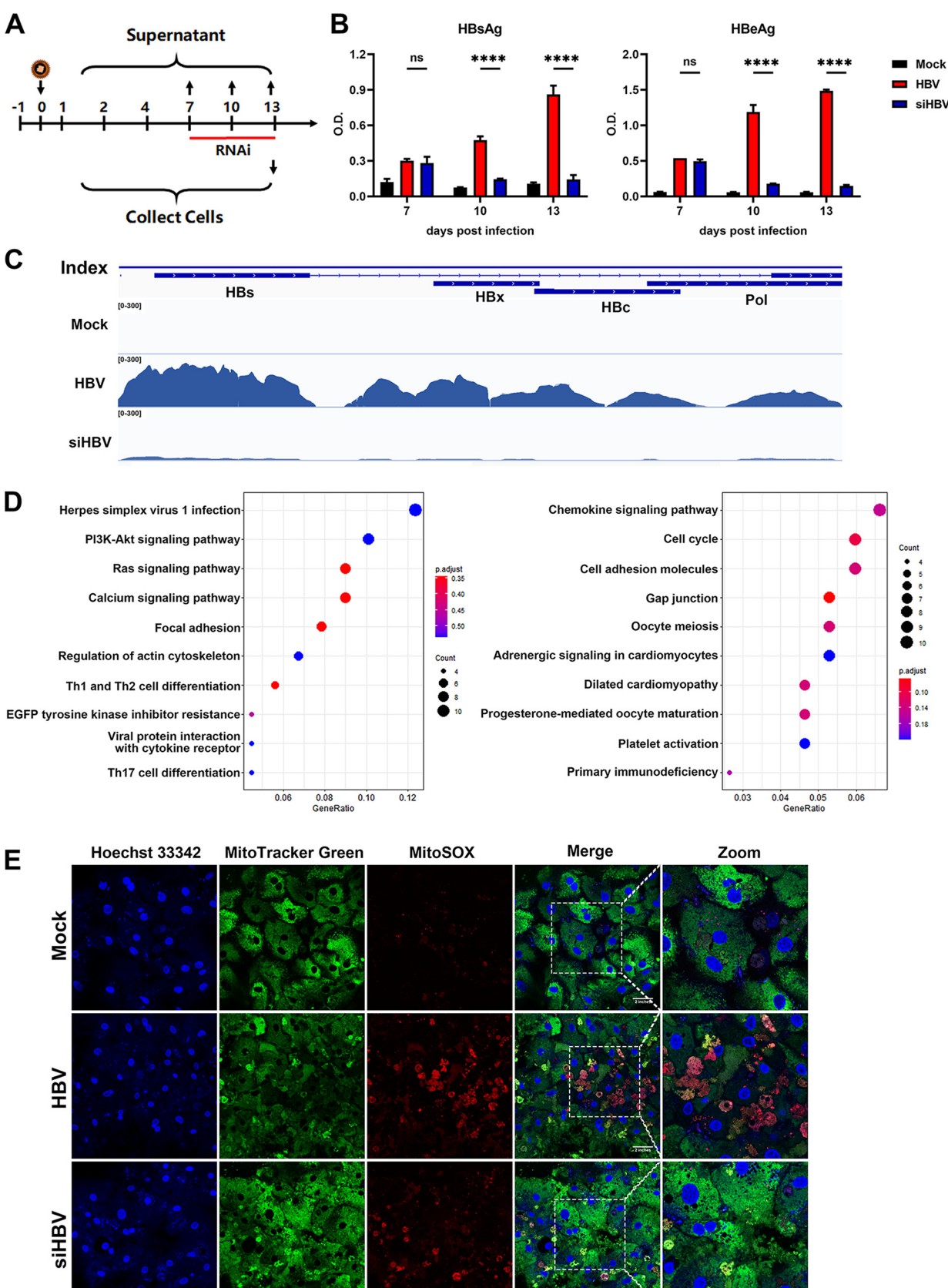

**FIG 6** Blocking HBV antigen by RNAi treatment significantly relieved HBV-induced host remodeling. (A) 5C-PHH cells infected with HBV (MOI of 200) were treated with (siHBV) or without (HBV) a combination of two siRNAs targeting all HBV-derived RNAs (100 nM) at 7 dpi for 1 week.

using similar ways as reported previously. Chimeric reads mapping to the human and HBV genome were readily found in samples infected with HBV but not in mock-infected cells, and the numbers of chimeric reads increased with time (Fig. S8B and C). The HBV integration might be associated with HBV-induced cytopathology and hepatocellular carcinoma development. However, considering the relatively low integration rate, it might not be the major reason for cellular pathology and host remodeling.

The cytopathic effects of HBV challenge the common notion that HBV infection is noncytopathogenic (32). Recent studies suggested that liver injury can be caused by highly accumulated viral products in infected hepatocytes, the extent of which correlates closely with HBV antigen level and viral replication activity (33–35). Inefficient virus release that leads to accumulation of high levels of viral replicative intermediates in infected cells can further lead to liver injury, as characterized by disrupted intracellular organelles, endoplasmic reticulum stress, and alteration of mitochondrial function (34, 36, 37). Cytopathic changes can also be caused by accumulation of viral proteins, especially HBsAg, overexpression of which can increase the cell's susceptibility to apoptosis (38). Clinical observations support that HBV might be cytopathic since HBV itself can induce hepatic injury in patients with immunosuppression (3, 30, 39). By using the uPA-SCID mouse-based human liver cell chimeric model, HBV was shown to be directly cytopathic in conditions of severe immune suppression (34). We then inquired whether blocking HBV replication by NAs could rescue HBV-induced cytopathic effects. The results suggested that early administration of ETV globally reduced HBV DNAs and antigens and thus rescued several dysregulated proteins caused by long-term HBV exposure, among which degraded proteins accounted for the largest proportion. Furthermore, reduced viral replication and antigens could relieve oxidative stress and mitochondrial dysfunction and improve the capacity of antigen processing and presentation in HBV-infected cells. Strikingly, later treatment of HBV-infected hepatocytes at stable phase (10 to 28 dpi) with either ETV or TDF rarely rescued dysregulated proteins. This suggests that the viral antigen expression, rather than viral replication, might mainly account for the cytopathic effects of HBV in hepatocytes. Further interrogation of the effects of RNAi confirmed that blocking viral antigen expression with RNAi could efficiently relieve long-term HBV infection-induced cytopathic effects and host remodeling. Notably, the situation in the clinic is much more complicated than that in the cell culture-based model; that is, in addition to the reduction of viral antigen levels, the reduction of viral load may also contribute to reducing the cytopathic effects, as it may associate with the decreased overall viral burden and improvement of the status of antiviral immune responses. Moreover, there has been controversy over the necessity of antiviral treatment for immunotolerant phase CHB patients, and the present study provides a new perspective for considering early NA treatment and reducing viral antigen load with RNAi in chronic hepatitis B patients (40, 41).

The advance of high-throughput sequencing and quantitative proteomics has also provided insights into viral hijacking and host antagonism, aside from cell adaptation and remodeling, thus providing comprehensive information on viral-host interplay (42–45). Overexpression screening was further applied to interrogate whether these dysregulated proteins could participate in the processes of viral life cycle and host defenses. We identified multiple candidate proteins that enrich host viral factors and antiviral factors. Five host factors, including UBE2V2, EDF1, WDR3, ZFR, and ESRRB, were identified to significantly promote HBV antigen expression, while eight host factors, including PGAM5, EXOSC6, CAV1, CAV2, FUBP3, TRIM47, GET4, and HNRNPD, were identified to inhibit HBV antigen secretion. We further confirmed that these genes can

**FIG 6** Legend (Continued)
(B) Supernatants were collected every 3 days, and HBsAg and HBeAg analysis was applied. (C) Distribution of HBV transcripts along the HBV genome was quantified by nucleotide mapping. Read density for each track was represented by height on the y axis. (D) KEGG pathway enrichment terms enriched among genes rescued by RNAi therapy (left, cells downregulated by HBV while they were rescued by RNAi; right, cells upregulated by HBV while they were relieved by RNAi) are displayed. (E) Immunofluorescent analysis of Hoechst 33342 (blue), MitoTracker Green (green), and MitoSOX (red) stain in mock- and HBV-infected 5C-PHH cells with or without RNAi treatment, as indicated.

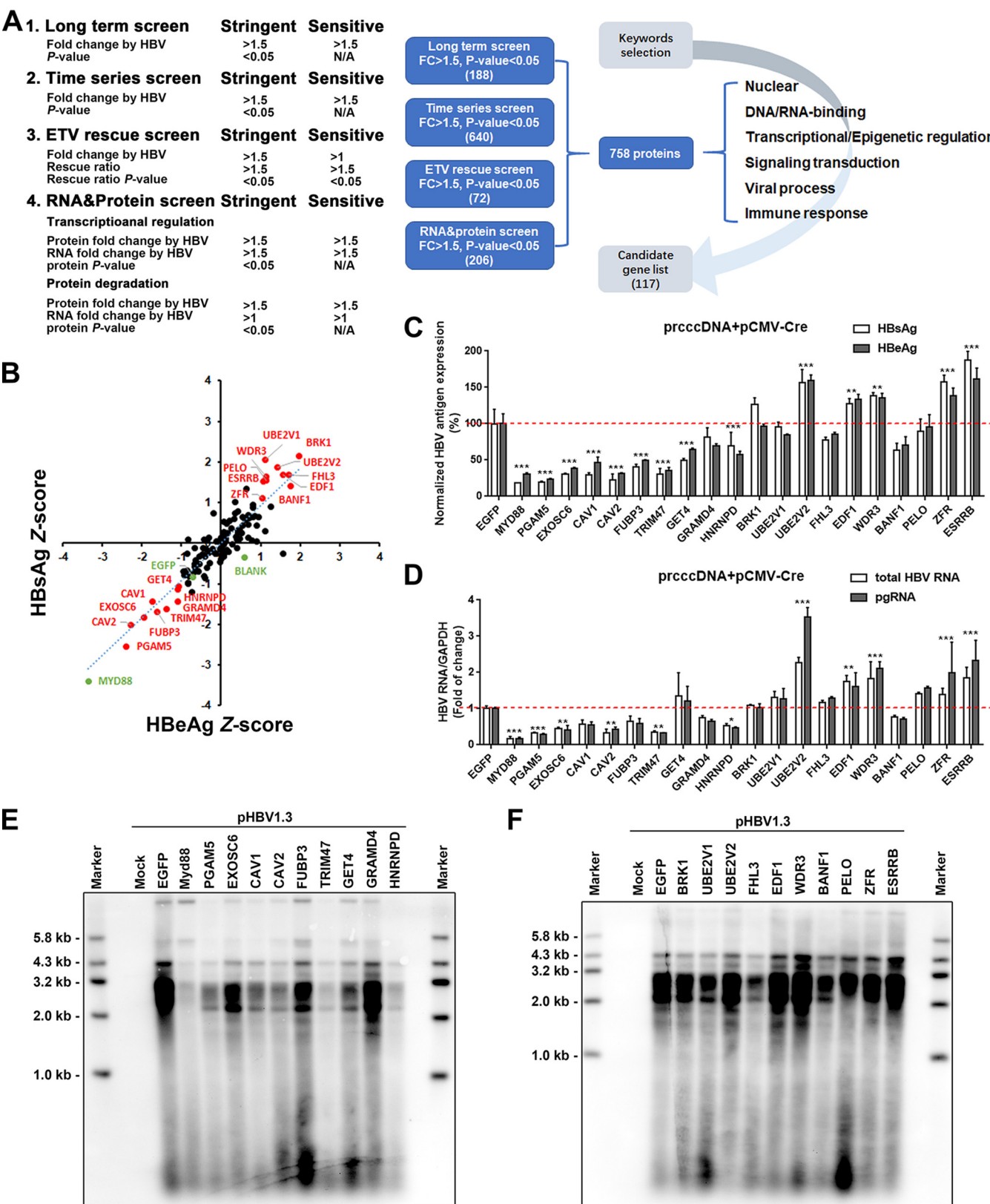

**FIG 7** Overexpression screening of dysregulated proteins identified in combined analysis. (A) Schematics of experiment workflow. Summary of sensitive and stringent criteria used in this study; this facilitated identification of a shortlist of hits dysregulated by HBV infection with high confidence. (B) HepG2 cells were cotransfected with overexpressed plasmids along with prcccDNA and pCMV-Cre plasmids. Dot plots showing the *Z*-score normalized average HBsAg and HBeAg values from two independent experiments. Top and bottom proteins are indicated by red dots. (C and D) Confirmation analysis for selected pro- and anti-HBV genes. MyD88 was included as a positive control. HBV antigen levels (C) and HBV RNA levels (D) were normalized to the enhanced green fluorescent protein (EGFP) control. (E and F) Selected antiviral (E) and proviral (F) gene expression plasmids were cotransfected with pHBV1.3 plasmid into HepG2 cells. Cells were harvested at 5 days posttransfection. Core particle DNA was detected by Southern blot analysis.

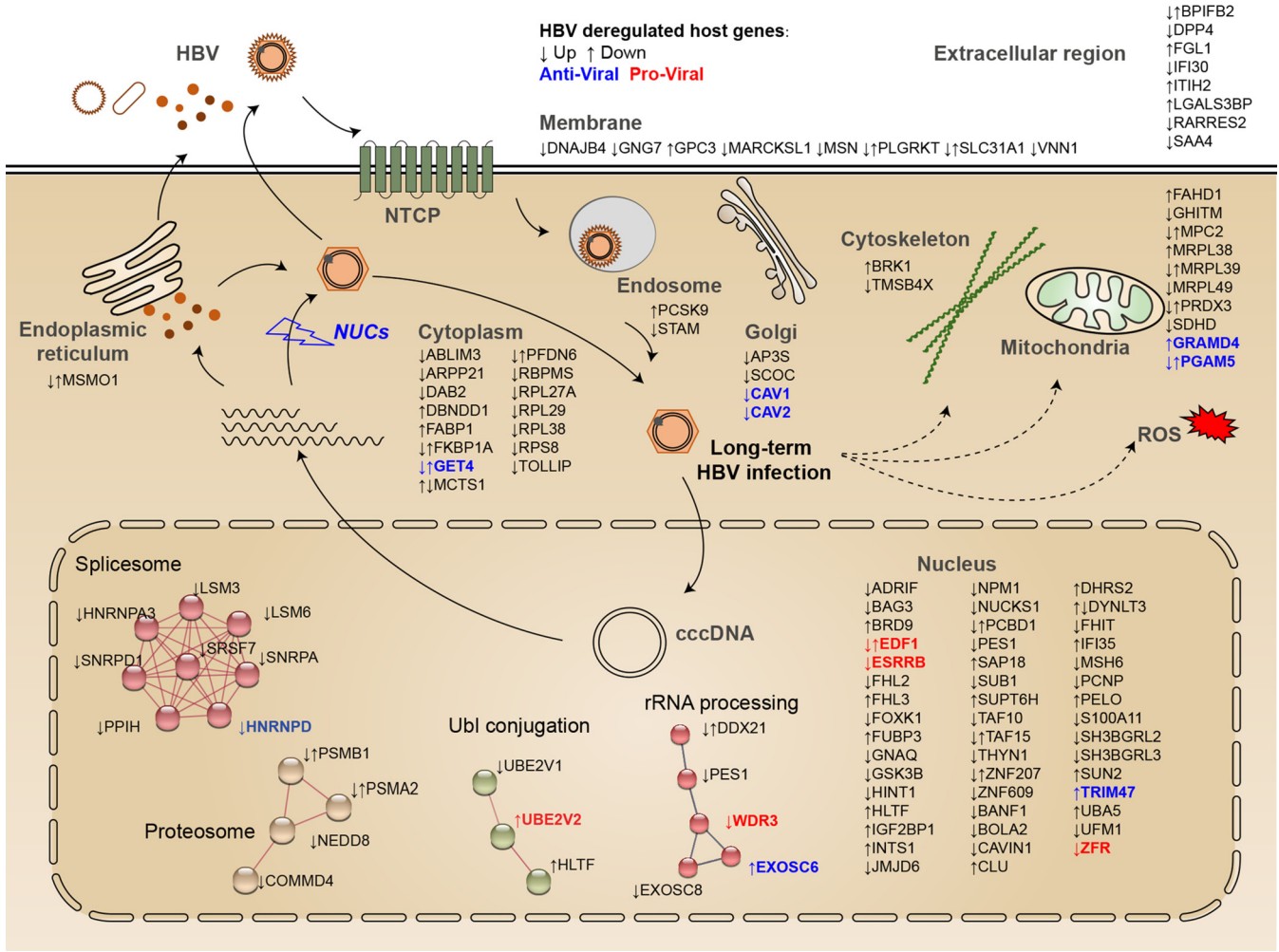

**FIG 8** Schematics illustrating the correlations between HBV and host cells. Global temporal quantitative proteomics analysis of long-term HBV-infected primary human hepatocytes accompanied by parallel transcriptomic analysis uncovered extensive remodeling of host proteome and transcriptome, as well as revealed cytopathic effects of long-term viral replication. Metabolic-, complement-, cytoskeleton-, mitochondrial-, and oxidation- related pathways were modulated at transcriptional or posttranscriptional levels, which could be partially rescued by early, but not late, nucleot(s)ide analogs therapy and could be relieved by RNAi. Thirteen proteins, many without characterized functions in promoting or inhibiting HBV replication, were identified by our unbiased analysis and further overexpression screen.

influence HBV transcription and replication in various degrees, as evidenced by decreased or increased intracellular HBV RNAs (including pgRNA and total RNA), core particle DNA levels, and the ratio of pgRNA to cccDNA, an indicator of cccDNA transcriptional activity. These genes distribute at different cellular compartments, like nuclei, mitochondria, and endosomes, which correlates closely with HBV life cycle and viral cytopathology. Clinical expression levels of genes in HBV-infected or HBV-associated HCC patients' liver tissues retrieved from the Gene Expression Omnibus database supported the findings in this study that HBV infection may change the levels of CAV1, CAV2, TRIM47, UBE2V2, WDR3, and ZFR (43) (Fig. S7D), though more functional studies are required to verify the roles of these genes in viral pathogenesis and antiviral responses.

The batch of PHH used in this study has been validated to be stable and representable in our previous studies (17, 22, 44). However, we cannot exclude the possibility that the situation may differ to some extent based on the variability from donor to donor. Moreover, since polyethylene glycol (PEG)-precipitated HepAD38 cell supernatant, but not highly purified HBV virions, was applied as viral inoculum here, the potential impact of PEG and other components coprecipitated with HBV virions on the observations could not be fully excluded. Therefore, further validations and investigations using more natural models and clinical samples would be necessary in the future. Also,

since 5C itself might influence multiple cellular signaling pathways which are important for viral-host interactions and potentially influence the HBV replication cycle, we have used relevant mock infections as 5C treatment controls at each time point, and the results suggest that little cytotoxic effect is induced by the 5C *per se*. However, considering the impact of the 5C treatment *per se* on cellular pathways, there is concern that the measured cellular gene expression profiles might not reflect the natural responses of PHH to HBV infection. Another limitation is that, although we focus on viral-host interactions induced by long-term HBV infection, the early changes of host gene expression profiles might be lost considering that HBV-cell contact may lead to an immediate cellular response. Due to the limitation of the detection method and sensitivity and because some high-abundance proteins could crowd out the data, it is technically hard to detect some low-abundance viral and host proteins. Together, this study has deepened our understanding of how HBV orchestrates host factors for viral replication and how the host mobilizes intracellular antagonisms to defend against viral invasion. More investigations on clarifying specific antiviral and pathogenic mechanisms of certain genes or pathways may provide novel therapeutic targets for viral control and protection against virus-induced cytopathic effects.

## MATERIALS AND METHODS

**Viral infection.** Cryopreserved human hepatocytes were purchased from BioreclamationITV. The day after plating, 5C-PHHs were infected with concentrated HBV particles (subtype ayw) collected from HepAD38 supernatant at a multiplicity of infection (MOI) of 200 in null medium (Williams' medium E containing B27, GlutaMAX, and penicillin-streptomycin [PenStrep]) containing 4% PEG 8000. For mock infection groups, cells were treated with null medium containing 4% PEG 8000. Twenty hours later, the medium was removed, and cells were rinsed with phosphate-buffered saline (PBS) three times. Cells were then cultured with 5C medium (null medium supplemented with Forskolin (20 $\mu$M), SB431542 (10 $\mu$M), IWP2 (0.5 $\mu$M), DAPT (5 $\mu$M), and LDN193189 (0.1 $\mu$M) for the indicated time before harvesting and testing).

**Whole-cell lysate protein digestion.** Proteins were extracted with lysis buffer (7 M urea, 4% SDS, 30 mM HEPES, 1 mM phenylmethylsulfonyl fluoride [PMSF], 2 mM EDTA, 10 mM dithiothreitol [DTT], 1$\times$ protease inhibitor cocktail) and then sonicated on ice and centrifuged at 13,000 rpm for 10 min at 4°C. The supernatants were then transferred to new tubes. Protein concentrations were determined by bicinchoninic acid (BCA) protein assay, and 100 $\mu$g protein per condition was transferred to new tubes and adjusted to a final volume of 100 $\mu$L with 100 mM triethylammonium bicarbonate (TEAB). We added 5 $\mu$L of 200-mM DTT, the samples were incubated at 55°C for 1 h; then, 5 $\mu$L of 770 mM iodoacetamide was added to the samples, and they were incubated in the dark for 30 min at room temperature. For each sample, proteins were precipitated with ice-cold acetone and then redissolved in 100 $\mu$L TEAB. Proteins were then digested with sequence-graded modified trypsin (Promega, Madison, WI), and the resultant peptide mixtures were labeled with iTRAQ 8-plex reagents (AB Sciex) or TMT 10-plex reagents (Thermo Scientific) according to the manufacturers' instructions, as described in the supplemental materials and methods. The labeled samples were combined, desalted using $C_{18}$ SPE column (Sep-Pak $C_{18}$, Waters, Milford, MA), and dried *in vacuo*.

**Liquid chromatography and mass spectrometry.** The Orbitrap Fusion mass spectrometer was operated in the data-dependent mode to switch automatically between mass spectrometry (MS) and tandem mass spectrometry (MS/MS) acquisition. Survey full-scan MS spectra (*m/z* 350 to 1,600) were acquired in Orbitrap with a mass resolution of 60,000 at *m/z* 200. The AGC target was set to 1,000,000, and the maximum injection time was 50 ms. MS/MS acquisition was performed in Orbitrap with a 3-s cycle time, and the resolution was 15,000 at *m/z* 200. The intensity threshold was 50,000, and the maximum injection time was 100 ms. The AGC target was set to 100,000, and the isolation window was 1.8 *m/z*. Ions with charge states of $2^+$, $3^+$, and $4^+$ were sequentially fragmented by higher-energy collisional dissociation (HCD) with a normalized collision energy (NCE) of 35%. In all cases, one microscan was recorded using dynamic exclusion of 20 s. MS/MS fixed first mass was set at 110.

**Data processing and protein identification.** Raw mass spectrometry data were extracted by Proteome Discoverer software (Thermo Fisher Scientific; version 1.4.0.288). Charge state deconvolution and deisotoping were not performed. Mascot (Matrix Science, London, UK; version 2.3) was set up to search the combined database, consisting of (i) human UniProt, and (ii) HBV UniProt, assuming the digestion enzyme trypsin. The percolator algorithm was used to control peptide-level false-discovery rates (FDR) lower than 1%. Only unique peptides were used for protein quantification, proteins contained at least two unique peptides, the method of normalization on the protein median was used to correct experimental bias, and the minimum number of proteins that must be observed was set to 1,000. The Orbitrap Fusion mass spectrometer was operated for mass spectrometry.

For protein quantification, reverse and contaminant proteins were removed, and then each reporter ion channel was summed across all quantified proteins and normalized assuming equal protein loading across samples. Hierachical centroid clustering based on uncentered Pearson correlation was performed using Cluster 3.0 (Stanford University) and visualized using Java TreeView (http://jtreeview.sourceforge.net/). K-means clustering

was performed using XLSTAT (Addinsoft), and each cluster was then subjected to hierarchical clustering using Cluster 3.0. The one-way analysis of variance (ANOVA) test was used to identify proteins differentially expressed. The adjusted *P*-value by Benjamini-Hochberg correlation for multiple testing was performed by Perseus (https://maxquant.net/perseus/) (46). The Benjamini-Hochberg-corrected *P*-value of <0.05 was considered statistically significant.

The Database for Annotation, Visualization and Integrated Discovery (DAVID) v6.8 was used to determine the enrichment of Gene Ontology "biological process" and "molecular function" terms and KEGG pathway enrichment (https://david.ncifcrf.gov/) (25). A background of all protein quantified within the relevant experiment was used. The Search Tool for the Retrieval of Interacting Genes/Proteins (STRING) v11.0 database was used to identify known protein interaction (https://string-db.org/). The Ingenuity Pathway Analysis (IPA) database was used to identify biological clusters and protein interaction networks.

**RNA-seq analysis.** RNA-seq analysis was performed in biological triplicate at three time points of infection, 2, 7, and 28 dpi. RNA was extracted at indicated time points, and poly(A) RNA was enriched for cDNA library construction. The libraries were constructed using the Illumina mRNA-Seq prep kit (Illumina), quantified by Agilent Bioanalyzer 2100 (Agilent Technology, Santa Clara, CA), and then pooled for Illumina sequencing. The sequencing reads from all RNA-seq experiments were trimmed by Cutadapt v3.1 (https://cutadapt.readthedocs.io/en/stable/) and then aligned to the human (hg19) reference genome and HBV genotype D genome (GenBank accession no. X02496.1) by Hisat2 (https://daehwankimlab.github.io/hisat2/) (47). The resulting files were sorted by SAMtools (http://samtools.sourceforge.net/) and analyzed for expression using featureCounts (http://subread.sourceforge.net/). For nucleoside mapping, HBV-derived reads were visualized by the IGV genome browser, and read counts per million total reads (read per million [RPM]) were calculated. Differential gene expression analyses were performed using DEseq2 (https://bioconductor.org/packages/DESeq2/) using fold change of ≥2 and *P* value of ≤0.05 as the cutoff.

**Transmission electron microscopy.** For transmission electron microscopy detection, samples were collected as previously described (48). Cells were then sliced and detected by JEM 1410 transmission electron microscope (JEOL, MA).

**Immunofluorescence microscopy.** Cells were washed with PBS twice and then incubated with MitoTracker Green (total mitochondrial mass) and MitoSOX (mitochondrial ROS) stain or stained with JC-1 at 37°C for 20 min, according to the manufacturer's protocol. Cell nuclei were stained with Hoechst 33342. Fluorescence was then observed using a confocal microscope (Zeiss LSM 7100).

**Quantitative reverse transcription-PCR.** Reverse transcription was performed using PrimeScript reverse transcription (RT) reagent kit with gDNA Eraser (TaKaRa; catalog no. RR047A). SYBR green reagents were utilized for quantitative PCR. Total RNAs were extracted and detected separately using specific primers as follows: for HBV pgRNA, forward, GCCTTAGAGTCTCCTGAGCA, and reverse, GAGGG AGTTCTTCTTCTAGG; for HBV total RNA, forward, GCTTTCACTTTCTCGCCAAC, and reverse, GAGTTCCGCA GTATGGATCG; and normalized to genomic *gapdh* (glyceraldehyde-3-phosphate dehydrogenase), forward, GGTATCGTGGAAGGACTCATGA, and reverse, ATGCCAGTGAGCTTCCCGTTCAGC.

**Detection of HBV antigens and HBV DNA.** HBV surface antigen (HBsAg) and HBeAg levels in the supernatant were measured using enzyme-linked immunosorbent assay (ELISA) (Kehua, Shanghai, China). HBV DNA levels in the medium were determined by qPCR using commercial kits (Shengxiang, Human, China).

**HBV core particle DNA extraction.** Cells were washed with PBS and lysed with 1 mL buffer A (10 mM Tris-HCl [pH 8.0], 50 mM NaCl, 1 mM EDTA, and 1% NP-40) at 4°C for 10 min. Samples were then transferred to new tubes and centrifuged at 13,000 rpm for 5 min at 4°C. The supernatants were collected, supplied with 10 mM $MgCl_2$, 100 $\mu$g/mL DNase I, incubated at 37°C for 1 h, and stopped with 20 mM EDTA. Core particles were then predicted with 7% PEG 8000 at 4°C overnight, centrifuged at 13,000 rpm for 10 min at 4°C, and resuspended with 500 $\mu$L buffer PK (10 mM Tris-HCl [pH 8.0], 100 mM NaCl, 1 mM EDTA, 1% SDS, and 0.5 mg/mL proteinase K) and incubated at 56°C for 2 h. Samples were then purified by phenol-chloroform (1:1) extraction and ethanol precipitation. Samples were then detected by Southern blot analysis.

**Statistical analysis.** GraphPad Prism was used to plot data and to perform statistical analysis. All the bar graphs were shown with the means ± standard deviations. Unpaired, two-tailed *t* test was used to estimate *P*-values unless otherwise specified.

**Data availability.** The data that support the findings of this study are available within the paper and supplemental material. The mass spectrometry proteomics data reported in this paper have been deposited to the ProteomeXchange Consortium (http://proteomecentral.proteomexchange.org) via the iProX (www.iprox.org) partner repository with the data set identifier PXD028190 (accession numbers IPX0003431000/IPX0003431001 and IPX004007000/IPX004007001) (49). RNA-seq metadata, processed data, and FASTQ files can be obtained from the Gene Expression Omnibus (GEO) repository (accession numbers GSE183156 and GSE194007).

## SUPPLEMENTAL MATERIAL

Supplemental material is available online only.
**SUPPLEMENTAL FILE 1**, XLSX file, 6 MB.
**SUPPLEMENTAL FILE 2**, XLSX file, 0.2 MB.
**SUPPLEMENTAL FILE 3**, XLSX file, 0.6 MB.
**SUPPLEMENTAL FILE 4**, XLSX file, 2.7 MB.
**SUPPLEMENTAL FILE 5**, XLSX file, 0.04 MB.
**SUPPLEMENTAL FILE 6**, PDF file, 3.5 MB.

## ACKNOWLEDGMENTS

This work was supported by the grants from the National Natural Science Foundation of China (91842309, 81974304, and 82022043), the National Key R&D Program of China (2021YFC2300600), the Shanghai Rising-Star Program (20QA1400700), Shanghai Municipal Education Commission (201701070007E00057), the CAMS Innovation Fund for Medical Sciences (2019-12M-5-040), the Major Special Projects of Basic Research of Shanghai Science and Technology Commission (18JC1411100), and the Local Innovative and Research Teams Project of Guangdong Pearl River Talents Program (2017BT01S131).

W.Z., K.H., and J.Y. performed the experiments with contributions from J.D., C.H., Y.L., Z.F., M.W., and C.W. J.C. designed the study and wrote the manuscript, and Z.Y. supported and supervised the study.

All authors who have taken part in this study declare that we do not have anything to disclose regarding funding or conflict of interest with respect to the manuscript.

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
