## [Reviewer comments · Microbiology Spectrum]

Microbiology Spectrum

Long-Term Hepatitis B Virus Infection Induces Cytopathic Effects in Primary Human Hepatocytes, and Can be Partially Reversed by Antiviral Therapy

Zhenghong Yuan, Wenjing Zai, Kongying Hu, Jianyu Ye, Jiahui Ding, Chao Huang, Yaming Li, Zhong Fang, Min Wu, Cong Wang, and Jieliang Chen

Corresponding Author(s): Zhenghong Yuan, Key Laboratory of Medical Molecular Virology (MOE/NHC/CAMS), School of Basic Medical Sciences, Shanghai Medical College, Fudan University, Shanghai, China

Review Timeline:

Submission Date:	September 10, 2021
Editorial Decision:	December 1, 2021
Revision Received:	January 21, 2022
Accepted:	January 21, 2022

Editor: Leiliang Zhang

Reviewer(s): Disclosure of reviewer identity is with reference to reviewer comments included in decision letter(s). The following individuals involved in review of your submission have agreed to reveal their identity: Mengji Lu (Reviewer #1)

Transaction Report:

DOI: <https://doi.org/10.1128/spectrum.01328-21>

December 1, 2021

Prof. Zhenghong Yuan
Fudan University
Department of Medical Microbiology
138 YiXueYuan Road
Shanghai 200032
China

Re: Spectrum01328-21 (Long-Term Hepatitis B Virus Infection Induces Cytopathic Effects in Primary Human Hepatocytes, and Can be Partially Reversed by Antiviral Therapy)

Dear Prof. Zhenghong Yuan:

Link Not Available

Sincerely,

Leiliang Zhang

Journals Department
Reviewer comments:

Reviewer #1 (Comments for the Author):

Zai et al. performed proteome and transcriptome analysis for HBV infection of 5 chemicals-cultured primary human hepatocytes. They found that HBV infection of 5C-PHH led to changes in the protein and RNA expression and suggested that long term HBV infection may cause cytopathic effects. Antiviral treatment using nucleoside analogues reduced the HBV infection-mediated changes of cellular gene expression. Finally, a screening of 117 selected genes demonstrated that a part of those genes may have inhibit or promote HBV gene expression and replication.

Overall, this is an interesting study with a lot of useful data. The identified candidate genes relevant for HBV replication may be functionally examined in the future studies.

The major limitation of this study

1. As PHHs are treatment with 5 chemicals that block major cellular pathways, the measured cellular gene expression profiles do not reflect the natural responses of PHHs to HBV exposure and infection. The authors should be aware of this limitation and discuss it.

2. The first sampling time point was day 2. As HBV-cell contact may lead to an immediate cellular response, the early changes of host gene expression profiles may be missed in this study.
3. It is not noted what a percentage of PHHs was infected during the culturing?
4. Verification of the proteome and transcriptome data by western blotting and real time RT-PCRs for selected genes could strength the conclusion.

Reviewer #2 (Comments for the Author):

In general, HBV-infection is considered non-cytopathogenic. However, this manuscript challenged this common notion by demonstrated that long-term HBV infection induces cytopathic effects in primary human hepatocytes under their 5C-PHH culture system. Though the discovery is of interesting, there are several aspects need to be cleared which would further support their conclusions.

1. Regarding to the 5C-PHH culture system, some more information relevant to cell's hepatic characteristics of the cultured PHH would be needed, after a long-term culture. Also, the proliferation status of the cells, since several recent papers has demonstrated that the cell cycle related genes expression would affect HBV cccDNA transcription and viral replication.

2. Instead NAs, use of RNAi to block viral protein production could be more helpful to support authors' speculation that long term HBV exposure caused host proteins dysregularization and cytopathic effects. Since as mentioned by the authors in the discussion that the viral antigen expression rather than viral replication might mainly account for the cytopathic effects of HBV in hepatocytes.

3. There are more than 5000 host proteins has been quantified. In contrast, only one viral protein (HBc) included. How about p22, p17, HBx and HBsAg? and why?

Finally, how about the integration of HBV DNA in this long term infection model? This would an interesting quetion, though it is not directly associated with the current topic of this manuscript.

Staff Comments:

Preparing Revision Guidelines

Please return the manuscript within 60 days; if you cannot complete the modification within this time period, please contact me. If you do not wish to modify the manuscript and prefer to submit it to another journal, please notify me of your decision immediately so that the manuscript may be formally withdrawn from consideration by Microbiology Spectrum.

Dear Editor,

Thank you very much for your letter and suggestions on our manuscript entitled “Long-Term Hepatitis B Virus Infection Induces Cytopathic Effects in Primary Human Hepatocytes, and Can be Partially Reversed by Antiviral Therapy” (Manuscript No. **Spectrum01328-21**). According to the reviewer’s comments, we have revised our manuscript carefully with colored text and made point-by-point responses to the reviewers below this letter.

We deeply appreciate your consideration of the revised paper for publication in your journal and look forward to hearing from you at your earliest convenience.

Best wishes!

Sincerely yours,

Zhenghong Yuan, M.D., Ph.D. (zhyuan@shmu.edu.cn)
Professor and Director, Key Laboratory of Medical Molecular Virology, Shanghai Medical College, Fudan University, Shanghai 200032, China.

Jieliang Chen, M.D., Ph.D. (jieliangchen@fudan.edu.cn)
A/Professor, Key Laboratory of Medical Molecular Virology, Shanghai Medical College, Fudan University, Shanghai 200032, China.

Point-by-point response to the reviewers' comments

Reviewer #1 (Comments for the Author):

Zai et al. performed proteome and transcriptome analysis for HBV infection of 5 chemicals-cultured primary human hepatocytes. They found that HBV infection of 5C-PHH led to changes in the protein and RNA expression and suggested that long term HBV infection may cause cytopathic effects. Antiviral treatment using nucleoside analogues reduced the HBV infection-mediated changes of cellular gene expression. Finally, a screening of 117 selected genes demonstrated that a part of those genes may have inhibit or promote HBV gene expression and replication.

Overall, this is an interesting study with a lot of useful data. The identified candidate genes relevant for HBV replication may be functionally examined in the future studies.

The major limitation of this study

1. As PHHs are treatment with 5 chemicals that block major cellular pathways, the measured cellular gene expression profiles do not reflect the natural responses of PHHs to HBV exposure and infection. The authors should be aware of this limitation and discuss it.

Thanks for the comments! As the reviewer pointed out, the 5C culture might affect cellular signaling pathways, which potentially influence HBV replication cycle and HBV-host interaction, in addition to supporting the maintenance of hepatocyte function. To address this concern, we have used mock-infected controls during the whole period of 5C culture. By using this strategy, the signature of HBV infection at each time point that we observed could be mainly attributed to the result of HBV infection, rather than that of the 5C culture system.

However, we still could not avoid the potential limitations caused by 5 chemicals that block major cellular pathways as the reviewer suggested. In the revised manuscript, we have thus added some notes to the discussion section in the revised manuscript(p415): “considering the impact of the 5C-treatment *per se* on cellular pathways, there is concern that the measured cellular gene expression profiles might not fully reflect the natural responses of PHH to HBV infection.”

2. The first sampling time point was day 2. As HBV-cell contact may lead to an immediate cellular response, the early changes of host gene expression profiles may be missed in this study.

Thanks for the comments.

Previous studies of HBV-host interactions did focus mainly on relative early time points, since the traditional culture system could not support long term culture of primary human hepatocytes (Ancey PB, et al., *Oncotarget* 2015; Lamontagne J, et al., *PLoS Pathog* 2016; Xia Y, et al., *J Virol* 2018). However, as the 5C-PHH culture system enables long-term culture of PHH and efficient HBV infection, the present study focused on the changes induced by long-term HBV infection in this system, rather than the early time points (Xiang C, et al., *Science* 2019). Besides, the early

responses of PHH to HBV infection have been investigated previously (within 24 h), and the results showed that gene expression changes were highly dynamic and cumulative across the time-course experiment, independently of HBV infection (**Ancey PB, et al., Oncotarget 2015**). The transcriptomic analysis of AdGFP-HBV transduced PRH showed that the impact of HBV at early infection time point (24 h post infection) were majorly the “cell cycle” pathway, and later infection time (48 h post infection) impacted the metabolic processes (**Lamontagne J, et al., PLoS Pathog 2016**). These results were generally consistent with our observations in 5C-PHH cells at 2 dpi.

To address the reviewer’s concern, in the revised manuscript, we have added some notes to the discussion section as the reviewers suggested(p417): “Another limitation is that, although we focus on viral-host interactions induced by long-term HBV infection, the early changes of host gene expression profiles might be lost considering that HBV-cell contact may lead to an immediate cellular response.”

3. It is not noted what a percentage of PHHs was infected during the culturing?

Thanks for the comments.

As we previously reported, the efficiency of HBV infection has always been examined in each of our experiments using HBcAg immunostaining (**Sheng F, et al., Hepatology 2018; Chen J, et al., Hepatology 2021; Xiang C, et al., Science 2019**). In this study, we used the same batch of PHH, and confirmed that the 5C-PHH exhibited high-efficient HBV infection via immunofluorescence staining with anti-HBc antibody (> 80 %), the same as we previously reported (**Xiang C, et al., Science 2019**).

FIG S1 (B) 5C-PHH were infected with HBV at MOI of 200, and the efficiency of HBV infection were confirmed by immunofluorescence staining with Anti-HBc antibody.

The infection efficiency of the same batch of PHH has also been detected by FISH analysis established by our groups, and was published recently (**FIG. S6B, Yue, L, et al., PLoS pathogens 2021**). The probe was designed for visualizing minus-strand DNA [(-) DNA], and the molecular specificity of the signal was confirmed by pre-treatment with DNase I before. The labeled (-) DNA puncta were abundant in HBV-infected 5C-PHH, indicating successful and efficient HBV infection (> 80 %

infection efficiency) in our system.

(Yue, L, et al., PLoS pathogens 2021, doi: 10.1371/journal.ppat.1009838.)

FIG. S6B PHH and HBV-infected PHH were detected for (-) DNA and Smc6 (bottom panel). Scale bar, 4 μ m. Smc6-positive and (-) DNA-negative cells were indicated by solid white arrows and autofluorescence were indicated by thin white arrows (Inset of the bottom panel).

In the revised manuscript, descriptions about the infection efficiency were added(p111): “>80 % infection efficiency was achieved in this culture system supported by the result of immunofluorescence staining with anti-HBc antibody (Fig. S1B), as we previous observed (18, 22-24).”

4. Verification of the proteome and transcriptome data by western blotting and real time RT-PCRs for selected genes could strength the conclusion.

Thanks for the comments.

As the reviewers suggested, expression levels of relative genes referring to **Fig. S4D** were examined by Q-PCR analysis with specific primers, considering that these genes showed good protein-RNA correlations and pathway enrichment according our proteomics and transcriptomic database. The results below (**Fig. S8A**) indicated that the expression levels of genes associated with “complement and coagulation cascades” (C9, PLG), “metabolic pathways” (FABP1, PCSK9, HSD11B1) and “oxidation-reduction process” (ALDH1L1, ALDH2, CYP2E1, ME1) were upregulated, and the expression levels of genes related to “actin-filament binding” (CNN1, CNN2, FLNA, TPM4, TNSB4X) were downregulated, which were consistent with and strongly supported our proteomics and transcriptomics data.

FIG S8 (A) Expression levels of genes of mock- or HBV- infected 5C-PHH cells associated with **Figure S4D** were analyzed by Q-PCR with specific primers.

Besides, the expression levels of genes with antiviral properties, which were identified by our overexpression screening as shown in **Fig. 7**, were examined by Western blot analysis (**Fig. S7B**). The results showed that the expression levels of representative proteins including CAV1, CAV2, PGAM5 and TRIM47 in 5C system were highly consistent with the results of proteomics.

FIG S7 (B) Representative immunoblots of CAV1, CAV2, PGAM5 and TRIM47 and load-control β -actin of mock- or HBV- infected 5C-PHH cells at indicated time points.

Moreover, clinical expression levels of genes in HBV-infected or HBV-associated HCC patients liver tissues were re-analyzed based on the Gene Expression Omnibus database. The results also supported the findings in this study that HBV infection could change the levels of CAV1, CAV2, TRIM47, UBE2V2, WDR3 and ZFR (**Fig. S7D**).

FIG S7 (D) Expression levels of the indicated genes in HBV-infected patients with undetectable or detectable HBV DNA comparing to healthy patients (HBV DNA (-), n = 32, HBV DNA (+), n = 90); gene expression levels in HBV-infected patients at different stage of disease (tolerance, n = 22; clearance, n = 50; inactive, n = 11); and gene expression levels in tumors and adjacent tissues in HBV-associated HCC patients from two different cohorts (cohort1: non-tumor, n = 198, tumor, n = 98; cohort2: non-tumor, n = 5, tumor, n = 50). For more details, see “materials and methods”.

These results together suggested that the differential gene expression profiles based on our proteomics and transcriptomics datasets are reliable and can be validated by Q-PCR. More functional studies are required to verify the functions of specific genes and pathways in viral pathogenesis and antiviral responses in the future.

To address these concerns, in the revised manuscript, we have added the results of Q-PCR in the supplementary materials and added some notes to the discussion section as the reviewer suggested(p335): “The expression levels of genes associated with host remodeling and cytopathology induced by long-term HBV infection were verified by Q-PCR analysis, and the results showed high consistency with the proteomic and transcriptomic datasets (Fig. S8A), which further validated the reliability of our conclusions.”

Reviewer #2 (Comments for the Author):

In general, HBV-infection is considered non-cytopathogenic. However, this manuscript challenged this common notion by demonstrated that long-term HBV

infection induces cytopathic effects in primary human hepatocytes under their 5C-PHH culture system. Though the discovery is of interesting, there are several aspects need to be cleared which would further support their conclusions.

1. Regarding to the 5C-PHH culture system, some more information relevant to cell's hepatic characteristics of the cultured PHH would be needed, after a long-term culture. Also, the proliferation status of the cells, since several recent papers has demonstrated that the cell cycle related genes expression would affect HBV cccDNA transcription and viral replication.

Thanks for the comments.

The 5C-PHH culture system enables long-term observation of changes induced by persistent HBV-infection. It is indeed important to confirm the hepatic characteristics and proliferation status of the long-term cultured PHH, as the reviewers suggested, which might affect cccDNA transcription and viral replication.

To address these concerns, we have used our RNA-seq data to confirm the 5C-PHH's faithful recapitulation of liver-specific transcriptional profiles. Using a previous generated list of drug-metabolizing enzymes, we evaluated and confirmed the hepatic phenotypes of our system (**Shan J, et al., Nat Chem Biol 2013; Winer BY, et al., Hepatology 2020; Xiang C, et al., Science 2019**). The log₂-transformed counts for this panel of genes in each of our samples were displayed below, and the results showed that these genes remained largely unchanged independent of the time, infection, or treatment (**Fig. S1D**).

FIG S1 (D) Heatmaps displaying log₂-transformed gene expression profiles of drug-metabolizing, live-specific transcripts in 5C-PHH.

In addition, we analyzed the expression levels of the hepatic surrogate functional markers (ALB, BCRP, CAR, CEBPA, CREB3L3, CPS1, MRP2, NAG, NTCP) and the hepatic transcription factors (FOXA1, FOXA3, HNF1A, HNF4A, HNF6A, HNF6B, KLF15) in 5C-PHHs after long-term culture by Q-PCR analysis. The results shown below indicated that the expressions of hepatocyte-functional genes and key transcriptional factors were generally stable during the period of long-term 5C culture (**Fig. S1E**).

FIG S1 (E) Gene expression levels of hepatic surrogate functional markers and the hepatic transcription factors of mock- or HBV- infected 5C-PHH cells at the indicated time points were analyzed by Q-PCR with specific primers.

Similar observations have also been reported in our previous articles, which indicated that the drug metabolizing gene expression profiling in 5C-PHHs resembles that of non-cultured adult liver tissue and PHHs (Xiang C, et al., Science 2019, Fig. S12-14). Also, the expression levels of hepatocyte-functional markers were analyzed, the albumin secretion and urea synthesis of cultured 5C-PHH were detected, and hierarchical clustering of global gene expression profiles were determined previously (Xiang C, et al., Science 2019, Fig. 1C-E). These results together confirmed that 5C condition maintained the global gene expression pattern and hepatocyte characteristics of human hepatocytes over the long-term culture durations.

To address the concerns, in the revised manuscripts, we have included the results mentioned above as the supplementary materials and added some notes in the result section as the reviewers suggested(p119): “The gene expression profiles of liver specific drug-metabolizing enzymes were analyzed, and the expression levels of hepatic surrogate functional markers and hepatic transcriptional factors in 5C-PHHs were determined. The results together showed that the 5C condition effectively supported the expression of hepatic functional genes and maintained the characteristics of human hepatocytes over the long-term culture durations (Fig. S1D and E).”

2. Instead NAs, use of RNAi to block viral protein production could be more helpful to support authors' speculation that long term HBV exposure caused host proteins dysregulation and cytopathic effects. Since as mentioned by the authors in the discussion that the viral antigen expression rather than viral replication might mainly account for the cytopathic effects of HBV in hepatocytes.

Thanks for the comments! According to the reviewers' suggestion, we have designed an experiment to evaluate the influence of RNAi on the status of HBV-infected hepatocytes by RNA-seq, proteomics and immunofluorescent staining analysis. As indicated by the results **shown below**, we found that a combination of two siRNA that targets all HBV RNAs (targeting both HBs and HBx sequences) efficiently blocked HBV antigen expression (both HBsAg and HBeAg) and significantly downregulated HBV RNA levels as evidenced by the HBV nucleotide mapping analysis (**Fig. 6A to C**). As speculated, RNAi significantly rescued multiple HBV infection-induced dysregulated gene expression according to the similar criteria shown in **Fig. S5A** (stringent criteria: >1.5-fold change, rescue ratio >1.5, rescue ratio *P*-value <0.05), including genes related to virus infection responses ("Herpes simplex virus 1 infection"), cancer-related signaling ("PI3K-Akt/Ras signaling pathway") and cytoskeleton ("Regulation of actin cytoskeleton"), as well as reversed genes associated with "cell cycle" and "cell adhesion molecules" (**Fig. 6D**). Immunofluorescent staining showed that RNAi treatment efficiently relieved HBV-infection induced oxidative stress and mitochondrial dysfunction (**Fig. 6E**).

FIG 6. Blocking HBV antigen by RNAi treatment significantly relieved HBV-induced host remodeling.

(A) 5C-PHH cells infected with HBV (MOI 200) were treated with (siHBV) or without (HBV) a combination of two siRNA targeting all HBV-derived RNAs (100 nM) at 7 dpi for 1 week. (B) Supernatants were collected each 3 days and applied to HBsAg and HBeAg analysis. (C) Distribution of HBV transcripts along HBV genome were quantified by nucleotide mapping. Read density for each track was represented by height on the y axis. (D) KEGG pathway enrichment terms enriched among genes rescued by RNAi therapy (left: downregulated by HBV while rescued by RNAi; right: upregulated by HBV while relieved by RNAi) were displayed. (E)

Immunofluorescent analysis of Hoechst 33342 (blue), MitoTracker Green (green) and MitoSOX (red) stain in mock- and HBV-infected 5C-PHH cells with or without RNAi treatment, as indicated.

In addition, as shown by the proteomics data, we found that RNAi could partially rescue the expression of proteins that were dysregulated by HBV infection, with a rate of ~17.2% (21/122) in downregulated and ~5.4% (7/130) in upregulated proteins, according to the “stringent criteria” (>1.5-fold change, rescue ratio >1.5, fold change *P*-value <0.05, rescue ratio *P*-value <0.05) (**Fig. S6A to C**). DAVID analysis found that the rescued proteins were enriched in pathways related to “Complement activation”, and “Actin monomer”, which tend to be dysregulated by long-term HBV infection (**Fig. S6D**). Further analysis according to the “sensitive criteria” showed that RNAi therapy could also enhance some immune related pathway in treated cells, such as “Innate immune response in mucosa”, “Antibacteria humoral response” and “Defense response to Gram-positive bacterium” (**Fig. S6E**).

FIG S6, related to FIG 6. Blocking HBV antigen by RNAi treatment significantly relieved HBV-induced host remodeling.

(A) Volcano plot of proteins quantified in comparison between HBV-infected replicates and relative mock-infected cells, and (B) volcano plot of proteins between siHBV-treated and untreated HBV-infected cells. *P*-values were estimated using two-tailed *t*-test (red dots: upregulated; blue dots: downregulated; > 1.5-fold change and *P* < 0.05). (C) Rescue ratios of proteins identified that could be deregulated by HBV infection while rescued by siHBV treatment, according to the “stringent criteria” shown on the right. (D) DAVID analysis of pathway enrichment among proteins that were degraded by HBV infection while rescued by siHBV treatment. (E) DAVID analysis of pathway enrichment among proteins identified that could be rescued by siHBV treatment according to the “sensitive criteria” shown on the right.

Together, these results confirmed the speculation that it was viral protein expression rather than viral replication that mainly account for long-term HBV infection-induced cellular pathology and host remodeling.

In the revised manuscript, we add the results associated with RNAi (**Fig. 6 and Fig. S6**), and added some notes in the result and discussion sections as follows(p237):

“Blocking HBV antigen by RNAi treatment significantly relieved HBV-induced host remodeling.”

Since viral replication were speculated not to be responsible for HBV-mediated cytopathy, we further inquired whether blocking viral antigen expression by RNAi could reverse long-term HBV infection-induced cellular pathology and host remodeling. We utilized a combination of two siRNA (siHBV, 100 nM) to degrade all kinds of HBV RNAs (targeting both HBs and HBx sequences). The siHBV was transfected by commercial reagents at 7 days post HBV infection, and the viral antigens were significantly downregulated both 3 and 6 days later, and the intracellular HBV RNA levels were efficiently reduced as evidenced by nucleotide mapping (Fig. 6A to C). Proteomic and transcriptomic analysis was then applied to identify genes or pathways that could be rescued by RNAi (Table S4). According to the “stringent criteria” in Figure S5A, we found that genes related to virus infection responses (“Herpes simplex virus 1 infection”), cancer-related signaling (“PI3K-Akt/Ras signaling pathway”) and cytoskeleton (“Regulation of actin cytoskeleton”), as well as genes associated with “cell cycle” and “cellular adhesion molecules” were reversed by RNAi treatment in the transcriptomic dataset (Fig. 6D).

Similarly, the proteomic dataset showed that RNAi therapy rescued some proteins that were dysregulated by HBV infection, with a ratio of ~17.2% (21/122) in downregulated and ~5.4% (7/130) in upregulated proteins according to the “stringent criteria” (Fig. S6A to C). DAVID analysis found that the rescued proteins were enriched in pathways related to “Complement activation” and “Actin monomer” that tend to be dysregulated by long-term HBV infection (Fig. S6D). Further analysis according to the “sensitive criteria” showed that RNAi therapy also enhanced some immune related pathways in treated cells, such as “Innate immune response in mucosa”, “Antibacteria humoral response” and “Defense response to Gram-positive bacterium” (Fig. S6E). Besides, immunofluorescent staining showed that RNAi treatment efficiently relieved HBV-infection induced oxidative stress and mitochondrial dysfunction (Fig. 6E). Together, our results indicated that using antiviral agents like RNAi to block HBV antigen expression could benefit infected cells, not only by rescuing dysregulated proteins, but also by enhancing host antiviral responses.”

“Further interrogation of the effects of RNAi confirmed that blocking viral antigen expression with RNAi could efficiently relieve long-term HBV infection-induced cytopathic effects and host remodeling.”

3. There are more than 5000 host proteins has been quantified. In contrast, only one viral protein (HBc) included. How about p22, p17, HBx and HBsAg? and why?

Thanks for the comments.

As the reviewers mentioned above, only one viral protein (HBc) were detected by our proteomics analysis, while other viral proteins like HBx and HBsAg were not detected. The main reason was probably due to the relatively low intracellular levels of these proteins, with HBsAg being a secretory protein and HBx being a short-lived protein (~3 h) (Korniyev D, et al., J Virol 2019). The intracellular expression levels of HBs, HBx, POL, HBc, and the relative β -actin loading control in mock- and HBV-infected cells (28 dpi) were determined by Western Blot analysis, and the results confirmed that the levels of HBx, HBs and POL were relatively low, comparing to HBc (Fig. S1C).

FIG S1 (C) Expression levels of intracellular HBs, HBx, POL, HBc, and load-control β -actin of mock- or HBV- infected 5C-PHH cells at 28 dpi were determined by western blot analysis.

As for p22 and p17 (HBeAg), since the amino sequence of p22/p17 were similar with HBc (p21), the peptides detected by our proteomics data could not discriminate between these proteins (**shown below**). However, considering the similar reasons that the intracellular expression level of the secreting HBeAg is relatively low, the peptides detected might majorly come from HBc antigen.

Capsid protein OS=Hepatitis B virus genotype D subtype ayw (isolate France/Tiollais/1979)
 OX=490133 GN=C PE=1 SV=1 - [CAPSD_HBVD3]
 MDIDPYKEFG ATVELLSFLP SDFPFSVR **DL LDTASALYRE** ALESPEHCSP HHTALRQAIL CWGELMTLAT
 WVGVNLEDPA SR**DLVVSYVN TNMGLK**FRQL LWFHISCLTFGRETVIEWYLV SFGVWIR**TPP AYRPPNAPIL**
STLPETTVVR RGRSPRRRT PSPRRRSQS PRRRSQSRE SQC

External core antigen OS=Hepatitis B virus genotype D subtype ayw (isolate France/Tiollais/1979)
 OX=490133 GN=C PE=1 SV=1 - [HBEAG_HBVD3]
MQLFHLCLII SCSCPTVQAS KLCLGWLWGM DIDPYKEFGA TVELLSFLPS DFFPFSVR**DLL DTASALYRE**EA
 LESPEHCSPH HTALRQAILC WGELMTLATW VGVNLEDPAS R**DLVVSYVNT NMGLK**FRQLL WFHISCLTFG
 RETVIEWYLV SFGVWIR**TPPA YRPPNAPILS TLPETTVVR**R RGRSPRRRTP SPRRRRSQSP RRRRSQSRES QC

Similar phenomena were also found previously in host proteins. Due to the limitation of resolution and sensitivity of the detection method, and that some high-abundance proteins could crowd out the data, it is technically hard to detect some low-abundance proteins (like SMC5/6).

To address this concern, we have added the results and some notes in the discussion section in the revised manuscript(p420): “Due to the limitation of the detection method and sensitivity, and that some high-abundance proteins could crowd out the data, it is technically hard to detect some low-abundance viral and host proteins.”

4. Finally, how about the integration of HBV DNA in this long term infection model? This would an interesting question, though it is not directly associated with the current topic of this manuscript.

Thanks for the comments.

As noted by the reviewers, HBV integration has been considered as an event closely coinciding with the HBV life cycles. Previous studies using HepG2-NTCP cells, HepaRG cells exposed to HBV and woodchucks *de novo* infected with wide-type WHV found that HBV integrates as early as in 30 min post-infection, with an integration rate of > 1 per 10,000 cells (Chauhan R, et al., *Oncogenesis* 2017; Chauhan R, et al., *Cancer Genet* 2019; Thomas Tu, et al., *J Virol* 2018). Whereas the integration status of HBV DNA after long-term HBV infection has not been reported to our knowledge.

According to the reviewers’ suggestion, we have tried to analyze the events of HBV integration by analyzing the human-HBV chimeric reads within our RNA-seq data using similar methods as reported previously (Zhang L, et al., *Proc Natl Acad Sci U S A* 2021; Kazachenka A, et al., *Front Microbiol* 2021; Ringlander J, et al., *J Viral Hepat* 2020). The junctions between human and HBV sequences were visualized by Circos (<http://circos.ca/>). As expected, chimeric reads mapping to the human and HBV genome were readily found in samples infected with HBV but not in mock-infected cells, and the numbers of chimeric reads increased with time on (data shown as below). And the chimeric junctions revealed that HBV integration distributes randomly over the entire human genome.

FIG S8 (B) Human-HBV chimeric junctions were analyzed and visualized by Circos.

We further confirmed the chimeric reads by visual inspection with the UCSC BLAT tool, and aligned the chimeric reads to the HBV genome. Representative examples of human-HBV chimeric reads are **shown below**, the HBV and human genomic sequences are shown in **red** and **blue**, respectively. Overlapping sequences are shown in boxed **green** letters.

E00552:372:H3F2NCCX2:1:1120:16549:24286

AAAAATGTCATGTTCCCTCAAGTAGAGCAGCAAAGTTCTTGCTAATGAGT**GTCCC**C
AACCTCCAATCACTCACCAACCTCCTGTCCCTCCAATTGTCCTGGTTATCGCTGGA
TGTGTCTGCGGCGTTTTATCATCTTCCTCTTCATCCTGC

E00552:372:H3F2NCCX2:1:1111:23145:4386

ATTAAAGGTCTTTGTACTAGGAGGCTGTAGGCATAAATTGGTCTGCGCACCAGCAC
CATGCAACTTTTTACCTC**TGTCT**TTCCCCATGGCCTTGCAAGCCAGGGTGGCTTT
GCAGCTTGCTACTCACGTAAGC

E00491:444:H3C7GCCX2:2:1204:11403:37313

CAGCCGATGTCTGGAGCCTGGGCGTGGCGCTCTTCACCATGCTGGCCGGTAAAAT
TCAGAGAGTAACCCCATCTCTTTGTTTTGTTAGGGTTTAAATGTATACCCAAAGAC
AAAAGAAAATTGGTAAAAGCGGTAAAAAGGGACTCAAG

E00552:372:H3F2NCCX2:1:2117:25276:25411

AGCACGGGACCATGCCGAACCTGCATGACTACTGCTCAAGGAACCTCTATGTATCC
CTCCTGTTGCTGTACCAAACCTTCGGACGGAAATTGCACCTGTATTCCCAT**TCGC**
TTCTCCCGGTGTGGTGTCCGCTGTTATTCTGCCATC

FIG S8 (C) Representative examples of human-HBV chimeric reads. HBV- and human- derived reads were shown in red and blue, respectively. Overlapping sequence are shown in green.

These results together indicated that, in the 5C-PHH culture system, HBV integration occurred within 2 days of infection, and increased with time on. Thus, the present model might be utilized as a novel *in vitro* system to interrogate the mechanisms of HBV DNA integration. However, much more systemic analysis is required to reveal more details.

In the revised manuscript, we have added the results mentioned above in the supplementary material and added some notes about HBV integration and discussed as below(p339): “Further, the possibility of HBV integration was investigated by analyzing the human-HBV chimeric reads within our RNA-seq data using similar ways as reported previously. Chimeric reads mapping to the human and HBV genome were readily found in samples infected with HBV but not in mock-infected cells, and the numbers of chimeric reads increased with time on (Fig. S8B and C). The HBV integration might be associated with HBV-induced cytopathology and hepatocellular

carcinoma development. However, considering the relative low integration rate, it might not be the major reasons for cellular pathology and host remodeling.”

January 21, 2022

Prof. Zhenghong Yuan
Key Laboratory of Medical Molecular Virology (MOE/NHC/CAMS), School of Basic Medical Sciences, Shanghai Medical College, Fudan University, Shanghai, China
Department of Medical Microbiology
138 YiXueYuan Road
Shanghai 200032
China

Re: Spectrum01328-21R1 (Long-Term Hepatitis B Virus Infection Induces Cytopathic Effects in Primary Human Hepatocytes, and Can be Partially Reversed by Antiviral Therapy)

Dear Prof. Zhenghong Yuan:

Your manuscript has been accepted, and I am forwarding it to the ASM Journals Department for publication. You will be notified when your proofs are ready to be viewed.

Sincerely,

Leiliang Zhang
Editor, Microbiology Spectrum

Journals Department
Table S4: Accept
Fig. S1, Fig. S2, Fig. S3, Fig. S4, Fig. S5, Fig. S6, Fig. S7, Fig. S8: Accept
Table S1: Accept
Table S3: Accept
Table S5: Accept
Table S2: Accept